# PROMPT ENGINEERING A PROMPT ENGINEER

## ABSTRACT

Prompt engineering is a challenging yet crucial task for optimizing the performance of large language models (LLMs). It requires complex reasoning to examine the model's errors, hypothesize what is missing or misleading in the current prompt, and communicate the task with clarity. While recent works indicate that LLMs can be meta-prompted to perform automatic prompt engineering, their potentials may not be fully untapped due to the lack of sufficient guidance to elicit complex reasoning capabilities in LLMs in the meta-prompt. In this work, we investigate the problem of "prompt engineering a prompt engineer"—constructing a meta-prompt that more effectively guides LLMs to perform automatic prompt engineering. We introduce and analyze key components, such as a step-by-step reasoning template and context specification, which lead to improved performance. Our final method, named PE2, finds a prompt that outperforms "let's think step by step" by 6.3% on the MultiArith dataset and 3.1% on the GSM8K dataset. To demonstrate its versatility, we apply PE2 to the Instruction Induction benchmark, a suite of counterfactual tasks, and real-world problems such as intent classification and movie recommendation. In these settings, PE2 achieves strong performance and outperforms prior automatic prompt engineering baselines. Further, we show that PE2 makes meaningful and targeted prompt edits, amends erroneous or incomplete prompts, devises multi-step strategies, and presents non-trivial counterfactual reasoning abilities.

## 1 INTRODUCTION

Large language models (LLMs) are powerful tools for many natural language processing tasks when given the right prompts.[1] However, finding the optimal prompt can be challenging due to the model's sensitivity (Jiang et al., 2020; Zhao et al., 2021; Lu et al., 2022), often necessitating extensive manual trial-and-error efforts. Moreover, once an initial prompt is deployed into production, unforeseen edge cases may emerge, demanding more rounds of manual efforts to further refine the prompt. These challenges give rise to an emerging research field of automatic prompt engineering. Within this field, a notable line of methods involves leveraging the capabilities of LLMs themselves (Zhou et al., 2023b; Pryzant et al., 2023). Specifically, this entails meta-prompting LLMs with instructions such as "inspect the current prompt and a batch of examples, then propose a new prompt."

While these methods achieve impressive performance, a subsequent question arises: What makes a good meta-prompt for automatic prompt engineering? To answer this question, we connect two key observations: (1) Prompt engineering itself is a complex language task that requires deep reasoning: it involves closely examining the model's errors, hypothesizing what is missing or misleading in the current prompt, and communicating the task more clearly to the LLM. (2) Complex reasoning capabilities in LLMs can be elicited by prompting the model to "think step by step" (Wei et al., 2022; Kojima et al., 2022) and can be further improved by instructing them to reflect on their outputs (Madaan et al., 2023; Chen et al., 2023).

Bridging these two observations, in this work, we prompt engineer a prompt engineer—we aim to construct a meta-prompt that guide LLMs to perform prompt engineering more effectively (§3; Fig. 2). By reflecting on the limitations of existing methods and incorporating recent advances in prompting for complex reasoning, we introduce meta-prompt components such as a detailed two-

---

[1]The word "prompt" is often overloaded with multiple meanings in recent literature. In this paper, prompt refers to the task description (*e.g.*, "Translate English to French") or instruction (*e.g.*, "Let's think step by step").

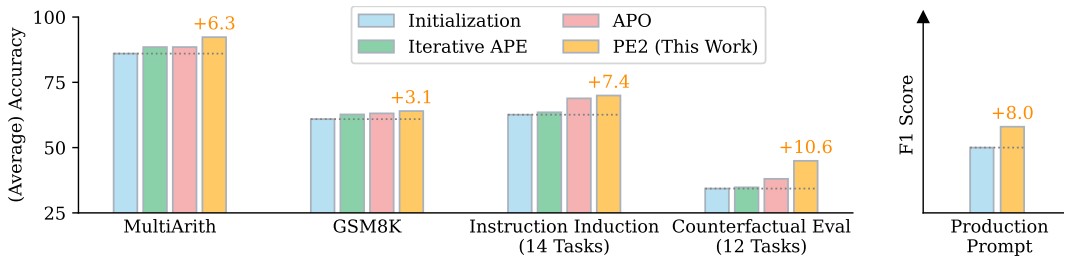

Figure 1: **Results Overview.** Our method PE2 consistently brings improvements over the prompt initialization (marked with orange text). It outperforms prompt optimization baselines Iterative APE (Zhou et al., 2023b) and APO (Pryzant et al., 2023). See full results on the Instruction Induction benchmark in Fig. 6, Counterfactual Eval in Fig. 7-8 and production prompt optimization in Fig. 12.

step task description, a step-by-step reasoning template and context specification, to explicitly guide the LLM to reason during the prompt engineering process.

Our final method, named PE2, achieves strong empirical performance (§5.1). When using TEXT-DAVINCI-003 as the task model, the prompts produced by PE2 surpass the zero-shot chain-of-thought prompt, "let's think step by step" (Kojima et al., 2022) by 6.3% on MultiArith and 3.1% on GSM8K. Moreover, PE2 outperforms two automatic prompt engineering baselines, Iterative APE (Zhou et al., 2023b) and APO (Pryzant et al., 2023) in multiple settings (Fig. 1). Notably, PE2 is most effective on counterfactual tasks (Wu et al., 2023), where the automatic prompt engineer is anticipated to reason about non-standard situations (*e.g.*, do addition in base-8 instead of base-10) and explain such situation to the task model through the prompt. Beyond academic datasets, PE2 proves its broad applicability in optimizing a lengthy, real-world prompt used in production.

Upon examining the prompt edit history of PE2 (§5.3), we find that PE2 consistently offers meaningful prompt edits (Table 4). It is able to amend erroneous or incomplete prompts and enrich the prompts with additional details, which leads to improved final performance. It is also able to devise multi-step plans for complex tasks. For example, in the task of movie recommendation, PE2 makes the plan to "consider factors such as genre, plot and style" in the prompt. Interestingly, when uninformed about performing addition in base-8, PE2 formulates its own arithmetic rules from the examples: "if both numbers are less than 50, add 2 to the sum. If either number is 50 or greater, add 22 to the sum." While this is an imperfect short-cut solution, it demonstrates PE2's non-trivial ability to reason in counterfactual situations. Despite these achievements, we also recognize the limitations and failure cases of PE2. We show that PE2 is influenced and bounded by the inherent limitations of current LLMs, such as neglecting given instructions and hallucinating incorrect rationales (Table 5).

## 2 BACKGROUND

In this section, we provide a formal formulation of the prompt engineering problem (§2.1), and describe a general framework of automatic prompt engineering using LLMs and meta-prompts (§2.2). Building on this foundation, in §3, we introduce the meta-prompt components and variants we investigate in this work.

### 2.1 PROMPT ENGINEERING

The goal of prompt engineering is to find the textual prompt $p^*$ that achieves the best performance on a given dataset $D$ when using a given LLM $\mathcal{M}_{task}$ as the task model. More specifically, we assume all datasets can be formatted as textual input-output pairs, *i.e.*, $D = \{(x, y)\}$. We are given a training set $D_{train}$ for optimizing the prompt, $D_{dev}$ for validation, and $D_{test}$ for final evaluation. Following the notations in Zhou et al. (2023b), the prompt engineering problem can be described as:

$$p^* = \arg\max_p \sum_{(x,y) \in D_{dev}} f(\mathcal{M}_{task}(x; p), y) \tag{1}$$

where $\mathcal{M}_{task}(x; p)$ is the output generated by the model when conditioning on the prompt $p$, and $f$ is a per-example evaluation function. For example, if the evaluation metric is exact match, $f(\mathcal{M}_{task}(x; p), y) = \mathbb{1}[\mathcal{M}_{task}(x; p) = y]$.

## 2.2 AUTOMATIC PROMPT ENGINEERING WITH LLMS

To alleviate the intensive efforts of human prompt engineering, recent works explore automating this process by meta-prompting LLMs to paraphrase the prompt (Zhou et al., 2023b) or refine the prompt by inspecting failure examples (Pryzant et al., 2023). In the following, we describe a framework that encapsulates these prior works and is employed in our investigation in later sections. It has three components: prompt initialization, new prompt proposal, and the search procedure.

**Prompt Initialization.** To start the prompt engineering process, a set of initial prompts $P^{(0)}$ is needed. We consider two initialization methods: **(1) Manual initialization** is applicable for tasks that has pre-existing prompts written by humans experts. For example, "Let's think step by step" leads to good performance on mathematical reasoning tasks and can be used as the initialization for prompt optimization. In **(2) Induction Initialization**, we follow the practice in Zhou et al. (2023b). We use a batch of examples $\{(x, y)\}$ from $D_{train}$ and a prompt $p^{init}$ ("Here are the input-output pairs. What is the instruction?"; See §B.1) to guide a LLM to generate a set of initial prompts $P^{(0)}$.

**New Prompt Proposal.** Given a set of initial prompts, the automatic prompt engineer will continuously propose new and potentially better prompts. At timestamp $t$, the prompt engineer is given a prompt $p^{(t)}$ and expected to write a new prompt $p^{(t+1)}$. Optionally, a batch of examples $B = \{(x, y, y')\}$ may be inspected in the new prompt proposal process. Here $y' = \mathcal{M}_{task}(x; p)$ represents model-generated output and $y$ represents the ground-truth label. We use $p^{meta}$ to denote a meta-prompt that is used to instruct the LLM $\mathcal{M}_{proposal}$ to propose new prompts. Therefore,

$$p^{(t+1)} = \mathcal{M}_{proposal}(p^{(t)}, B; p^{meta}) \tag{2}$$

Constructing a better meta-prompt $p^{meta}$ to improve the quality of the proposed prompt $p^{(t+1)}$ is the main focus of this study. We will describe multiple components and variants we consider in §3.

**Search Procedure.** As LLMs are sensitive to trivial prompt variations, it is possible that the newly proposed prompt $p^{(t+1)}$ under-performs the original prompt $p^{(t)}$. Therefore, automatic prompt engineering is typically combined with a back-tracking enabled search procedure. At timestamp $t$, we select $n$ best-performing prompts from *all* prompt candidates obtained in previous timestamps (*i.e.*, $P^{(0)} \cup P^{(1)} \cup ... \cup P^{(t)}$). For *each* of these $n$ prompts, we sample $m$ different batches $B$, and run the meta-prompt in Eq. 2 to produce $m$ new prompts. This results in $m \times n$ new prompts, which we denote as $P^{(t+1)}$ collectively and are used at the next timestamp $t + 1$. The prompt proposal and search procedure are described more formally in Algorithm 1 below.

---

**Algorithm 1** Search Procedure

1: $P^{(0)} = P_{init}$ or $P^{(0)} = \mathcal{M}_{init}(x_1, y_1, ..., x_n, y_n; p^{init})$      ▷ Manual init. or induction init.
2: **for** $t = 0, ..., T - 1$ **do**
3:      $P^{(t+1)} = \emptyset$
4:      **for** $p^{(t)} \in$ Select-Best$(\cup_{i=0}^{t} P^{(i)}, n)$ **do**      ▷ Select best $n$ prompts based on $D_{dev}$
5:          **for** $j = 1...m$ **do**
6:              $B =$ Sample$(D_{train})$      ▷ Sample a batch
7:              $p^{(t+1)} = \mathcal{M}_{optim}(p^{(t)}, B; p^{meta})$      ▷ New prompt proposal
8:              $P^{(t+1)} = P^{(t+1)} \cup \{p^{(t+1)}\}$
9:          **end for**
10:      **end for**
11: **end for**
12: **return** Select-Best$(\cup_{i=0}^{T} P^{(i)}, 1)$      ▷ Return the final best prompt based on $D_{dev}$

---

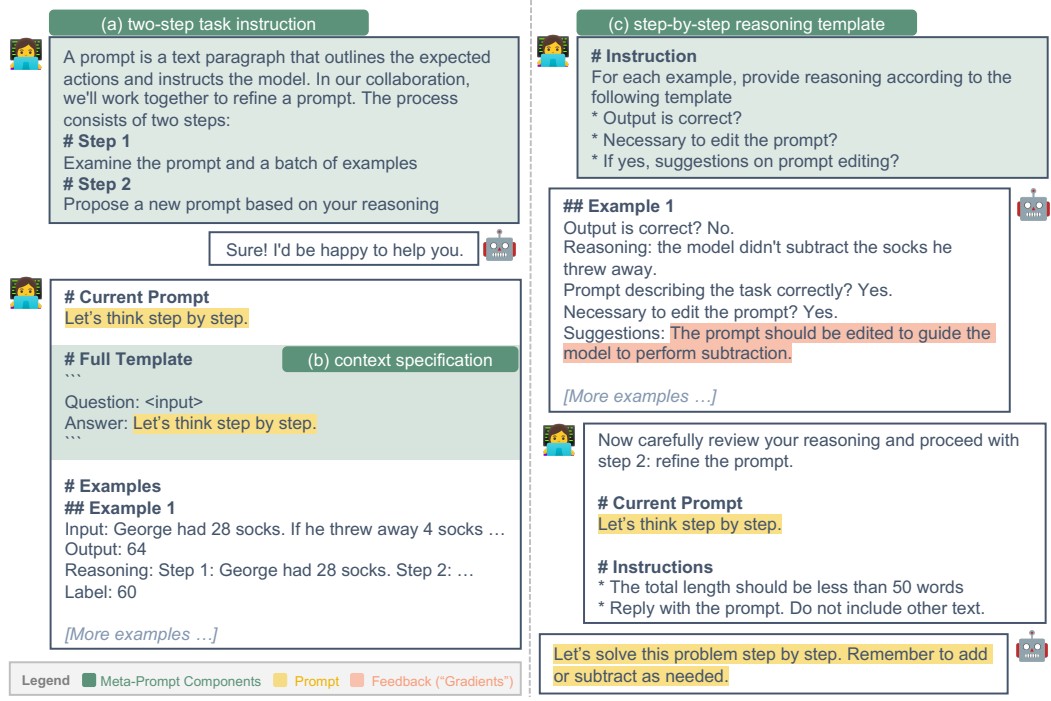

Figure 2: Illustration of the meta-prompt components. See §B.4 for the complete meta-prompt.

# 3 PROMPT ENGINEERING A PROMPT ENGINEER

Much like how the prompt plays an important role for the end task performance, the meta-prompt $p^{meta}$ introduced in Eq. 2 plays an important role in the quality of newly proposed prompts, and thus the overall quality of automatic prompt engineering. In this work, we focus on prompt engineering the meta-prompt $p^{meta}$—we develop meta-prompt components that can potentially help improve LLMs' prompt engineering quality.

In prior work, the meta-prompt either instructs the proposal model to generate prompts paraphrases (Zhou et al., 2023b) or contain minimal instructions about inspecting a batch of examples (Pryzant et al., 2023). As stated earlier, prompt engineering is a language task that requires complex reasoning, and these meta-prompts in prior works may not be sufficient to elicit the reasoning abilities. We posit that enriching the meta-prompt with additional instructions and context may be helpful. Therefore, we propose the following meta-prompt components, and visualize them in Fig. 2. We name our method using these three components as **PE2**, a prompt engineered prompt engineer.

(a) **Two-step Task Description.** The task of prompt engineering can be decomposed into two steps, as previously done in Pryzant et al. (2023): In step 1, the model is expected to inspect the current prompt and a batch. In step 2, the model is expected to compose an improved prompt.[2] However, in Pryzant et al. (2023) each step is explained *on the fly*. In contrast, we consider clarifying the two steps and communicating the expectations *upfront* in the meta-prompt.

(b) **Context Specification.** In practice, the location where the prompt is inserted in the whole input sequence is flexible. It may appear *before* the input text to describe the task, *e.g.*, "Translate English to French". It may appear *after* the input text, *e.g.*, "let's think step by step", to elicit reasoning capabilities. Recognizing these varying contexts, we explicitly specify the layout of the prompt and the input. For example, "Instruction: Translate English to French. Input: <input>" or "Q: <input> A: Let's think step by step."

---

[2]From the view of gradient descent, step 1 is analogous to computing the gradient or calling loss.backward(); and step 2 is analogous to applying the gradient or calling optimizer.step(). From the view of ReAct prompting (Yao et al., 2023), step 1 is reasoning and step 2 is acting.

(c) **Step-by-step Reasoning Template.** To encourage the model to examine *each* example in the batch $B$ closely and reflect on the limitations in the current prompt, we guide the prompt proposal model $\mathcal{M}_{proposal}$ to answer a list of questions. For example: Is the output correct? Is the prompt correctly describing the task? Is it necessary to edit the prompt? If yes, provide actionable suggestions on prompt editing.

**Other Meta-prompt Components We Tried.** Inspired by optimization concepts such as batch size, step size and momentum, we considered adding their verbalized counterparts to the meta-prompt and investigate their effects. We also considered adding prompt engineering tutorial in the meta-prompt to help the LLM better understand the task. However our observations on these components are mixed in the early stages of this work. Hence we describe and report them in Appendix A.2.

## 4 EXPERIMENT SETTING

### 4.1 TASKS

We use the following four groups of tasks to evaluate the effectiveness and limitations of PE2. More details (*e.g.*, dataset sizes, train-test splitting) are deferred in Appendix D.1.

**(1) Mathematical Reasoning.** We use MultiArith (Roy & Roth, 2015) and GSM8K (Cobbe et al., 2021), which contain grade school math problems that requires multiple steps of arithmetic operations. Previously, Kojima et al. (2022) discovered that "Let's think step by step" can elicit multi-step reasoning in LLMs to perform these two tasks. We use this prompt as the initialization.

**(2) Instruction Induction.** Instruction Induction (Honovich et al., 2023) is a benchmark for inferring the underlying instruction from few-shot examples. We use 14 selected tasks[3] that cover a wide range of use cases, *e.g.*, "Formality" is a task that aims at rephrasing a sentence in formal language; "Second Word Letter" aims at outputting the second letter in an input word. Full details on these tasks can be found in Table 12.

**(3) Counterfactual Evaluation.** We use the arithmetic, chess, and syntax tasks and their counterfactual variants introduced in Wu et al. (2023). For arithmetic, the original task is addition in base-10, and the counterfactual tasks are addition in base-8/9/11/16. We describe the chess and syntax tasks in Table 13. We use this set of tasks to observe whether PE2 can reason about counterfactual situations and communicate them to the task model.

**(4) Real-world Applications.** Lastly, we apply PE2 to optimize prompts for language tasks closely related to real-world applications. **(a)** Date understanding and movie recommendation from BIG-bench Hard (Suzgun et al., 2023). **(b)** A production prompt on a multi-label and hierarchical classification task. In this task, the model is expected to classify a natural language query into domain, and then into intents under the domain, and finally into a set of possible slots for each intent. The initial prompt has more than 5k tokens, and is carefully designed by experienced engineers.

### 4.2 EXPERIMENT DETAILS

**Compared Methods.** In addition to the multiple meta-prompt variants introduced in §3, we compare with the following three baselines. **(a) APE (Zhou et al., 2023b)**: The base version of APE is an initialization-only method and does not involve new prompt proposal steps. It uses an initialization prompt $p^{init}$ to generate multiple prompt candidates from a few examples, and select the best one among them based on $D_{dev}$ performance. **(b) Iterative APE (Zhou et al., 2023b)**: After initialization, $p^{meta}$ instructs the model to produce a paraphrase of $p^{(t)}$ and use it as $p^{(t+1)}$. **(c) APO (Pryzant et al., 2023)**: $p^{meta}$ contains minimal instructions on inspecting the batch $B$, generating textual "gradients" (feedback), and producing a new prompt $p^{(t+1)}$. We include the $p^{init}$ and $p^{meta}$ used in these baseline methods in Appendix B.

---

[3]To save computation, we removed 8 tasks since the baseline method APE already achieves near perfect accuracies (95%+) on them. We also removed 2 tasks due to their small dataset size ($\leq$ 50 examples). See Appendix D.1.

Table 1: Performance Comparison on Mathematical Reasoning Tasks. TD002/003 refers to text-davinci-002/003 models.

| Method | Task Model | Proposal Model | MultiArith Test | GSM8K Test |
|---|---|---|---|---|
| Fixed Prompt, Reported | | | | |
| Zero-shot CoT | TD002 | - | 78.7 | 40.7 |
| APE | TD002 | TD002 | 82.0 | 43.0 |
| Fixed Prompt, Reproduced | | | | |
| Zero-shot CoT | TD003 | - | 86.0 | 60.9 |
| APE | TD003 | - | 87.0 | 61.5 |
| Prompt Optimization | | | | |
| Iterative APE | TD003 | GPT-4 | 88.5 | 62.7 |
| APO | TD003 | GPT-4 | 88.5 | 63.1 |
| PE2 (this work) | TD003 | GPT-4 | **92.3** | **64.0** |

Table 2: Best prompts for MultiArith found by compared prompt optimization methods.

| Method | MultiArith Prompt |
|---|---|
| Fixed Prompt | |
| Zero-shot CoT | Let's think step by step. |
| APE | Let's work this out in a step by step way to be sure we have the right answer. |
| Prompt Optimization | |
| Iterative APE | Let's proceed in a methodical, step-by-step manner. |
| APO | Given the scenario, perform the necessary calculations step by step to find the final result. Consider all parts of the input and the sequence of events. |
| PE2 (this work) | Let's solve this problem by considering all the details. Pay attention to each piece of information, remember to add or subtract as needed, and perform the calculations step by step. |

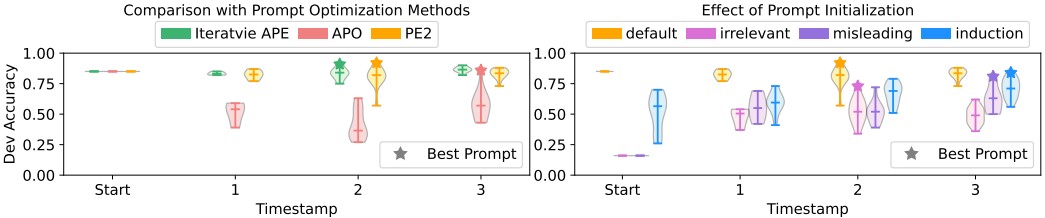

Figure 3: Prompt optimization dynamics on MultiArith. Left: Comparison with Iterative APE and APO. Right: Using different initializations.

**LLMs and Search Budget.** All the baselines mentioned above are encapsulated in the general framework introduced in §2.2. Due to cost and access considerations, we use GPT-4 (OpenAI, 2023) as prompt proposal model $\mathcal{M}_{proposal}$ and use TEXT-DAVINCI-003 (Ouyang et al., 2022) as the task model $\mathcal{M}_{task}$ performing the underlying task, for all compared methods. To ensure fair comparison, we use the same search budget for all prompt optimization methods. For experiments using induction initialization, 30 prompts are generated by $p^{init}$ and form the initial candidate set $P^{(0)}$. The number of optimization steps $T$ is set to be 3. At each timestamp, we select $n = 4$ best-performing prompts, and propose $m = 4$ prompts from each of them.

## 5 RESULTS AND ANALYSIS

### 5.1 MAIN RESULTS

**Improved baselines with more recent LLMs.** In Zero-shot CoT (Kojima et al., 2022) and APE (Zhou et al., 2023b), the results were obtained with a earlier TEXT-DAVINCI-002 model. We first rerun the prompts in these two works with TEXT-DAVINCI-003, a more recent model. In the top two sections in Table 1, we observe a significant performance boost by using TEXT-DAVINCI-003, suggesting that it is more capable of solving math reasoning problems with zero-shot CoT. Moreover, the gaps between the two prompts are narrowed (MultiArith: $3.3\% \rightarrow 1.0\%$, GSM8K: $2.3\% \rightarrow 0.6\%$), indicating TEXT-DAVINCI-003 has a reduced sensitivity to prompt paraphrasing. Given this, methods that rely on simple paraphrasing, such as Iterative APE, may not enhance the final outcome as effectively. More precise and targeted prompt edits are necessary to improve the performance.

**PE2 outperforms Iterative APE and APO on various tasks.** PE2 is able to find a prompt that achieves $92.3\%$ accuracy on MultiArith ($+6.3\%$ compared to Zero-shot CoT) and $64.0\%$ on GSM8K ($+3.1\%$). Additionally, prompts found by PE2 outperforms Iterative APE (Zhou et al., 2023b) and APO (Pryzant et al., 2023). In Fig. 1 we summarize performance gain obtained by PE2 on the instruction induction benchmark, counterfactual evaluation, and a production prompt. In Table 8 we report the results on date understanding and movie recommendation, two tasks from BIG-bench Hard. These results demonstrate that PE2 can be applied to diverse language tasks, and achieve strong performance. Notably, when induction initialization is used, PE2 outperforms APO on 11 out of 12 counterfactual tasks (Fig. 7), demonstrating that PE2 is capable of reasoning about contra-

dictions and counterfactual situations. We defer experiment details and performance breakdown for these benchmarks in Appendix A.3 and C.

**PE2 generates targeted prompt edits and high-quality prompts.** In Fig. 3(a) we plot the quality of prompt proposals over the course of prompt optimization. We observe very distinct patterns for the three prompt optimization methods: Iterative APE is based on paraphrasing, so the newly generated prompts have smaller variance. APO makes drastically large prompt edits and thus the performance drops in the first step. PE2 is the most stable one among the three methods. In Table 2, we list the optimal prompts found by these methods. Both APO and PE2 are able to provide instructions on "considering all parts / details". In addition, PE2 is designed to inspect the batch closely, enabling it to go beyond simple paraphrasing edits and make very specific prompt edits such as "remember to add or subtract as needed".

## 5.2 ABLATION STUDY

To demonstrate the effectiveness of the three meta-prompt components (two-step task description, step-by-step reasoning template, context specification) introduced in §3, we run ablation experiments by removing these components during prompt optimization on Multi-Arith and GSM8K. In these experiments, we make sure that the meta-prompt is still readable and contains minimal information about the task of prompt engineering. From the results in Table 3, we observe that these three components contribute significantly to prompt engineering quality. As shown in Fig. 4, the exclusion of any one of these components results in a wider variance in the quality distribution of newly-proposed prompts. Moreover, without these components, the proposal model more frequently suggests low-quality prompts compared to the default version.

We also conduct an ablation study on back-tracking (*i.e.*, at timestamp $t$, select top-performing prompts from $\cup_{i=0}^{t} P^{(i)}$ versus only $P^{(t)}$) and hard negative sampling (*i.e.*, the batch $B$ is sampled from the model's errors, versus the batch is randomly sampled from $D_{train}$). We conclude that these design choices are important in PE2 prompt optimization.

Table 3: Investigation on meta-prompt components and configurations.

| Method | MultiArith Dev | GSM8K Dev |
|---|---|---|
| PE2 (default) | 92.0 | 68.0 |
| Meta-prompt Components | | |
| - two-step task description | 89.0 | 66.0 |
| - step-by-step reasoning template | 87.0 | 61.0 |
| - context specification | 93.0 | 63.0 |
| Search Algorithm Configurations | | |
| - back-tracking | 90.0 | 66.0 |
| - hard negative sampling | 90.0 | 68.0 |

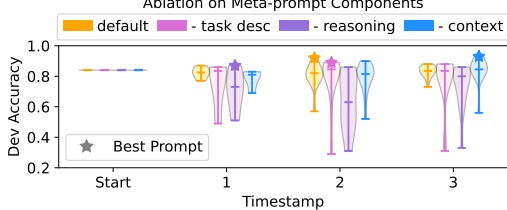

Figure 4: Prompt optimization dynamics on MultiArith when removing selected meta-prompt components. By removing one component, the new prompts have larger variance in their quality.

## 5.3 ANALYSIS AND CASE STUDY

**PE2 amends erroneous or incomplete instructions, and provides more details in instructions.** In Table 4 and Table 9, we present notable prompt edits made by PE2. In the task of rhymes (finding a word that rhymes with the input word), the initial prompt mistakenly suggests the task is about changing the first letter of a word. PE2 successfully correct this after one optimization step. We also find interesting prompt edits on the counterfactual tasks. In base-8 addition, when induction initialization is used (*i.e.*, the prompt engineer is uninformed with the information of base-8 and must infer it from the examples), PE2 is able to devise its own arithmetic rules (*e.g.*, add 22 to the sum) that is partially correct. Though this is an imperfect short-cut solution, it demonstrates PE2's ability to engage in sophisticated counterfactual reasoning.

**PE2 devises tailored multi-step plans for complex tasks.** As shown in Table 4, on the task of movie recommendation, PE2 is able to decompose the complex task into concrete criteria, such as genre, plot and actor, when determining movie similarities. On the task of date understanding, PE2

Table 4: Notable prompt edits made by PE2. See Table 9 for additional examples.

| Task | $t$ | Prompt | Dev Acc. |
|---|---|---|---|
| **Correct wrong or incomplete task instructions** | | | |
| Rhymes | 0 | Remove the first letter from each input word and then replace that first letter with a similar sounding letter or group of letters to form a new word. | 0.35 |
| | 1 | Generate a word that rhymes with the input word. | 0.45 |
| **Provide more specific context and details** | | | |
| Second Word Letter | 0 | Find the second letter in each word. | 0.9 |
| | 1 | Identify the second character in the provided word. | 0.95 |
| | 2 | Identify the second character from the start of the given word. | 1.0 |
| **Lay out tailored multi-step plans for complex problems** | | | |
| Movie Recommendation | 0 | Let's think step by step. | 0.58 |
| | 1 | Consider the genre, plot, and style of the input movies. Using this information, think step by step to identify which of the following options is most similar to the given movies. | 0.74 |
| | 2 | Considering factors such as genre, director, actors, release period, audience target, animation style, and humor, analyze the similarities among the given movies and identify the movie from the options that shares the most similarities. | 0.82 |
| Date Understanding | 0 | Let's think step by step. | 0.39 |
| | 2 | Analyzing the given information, let's calculate the solution. Remember to consider the context provided, such as references to 'today' or specific dates. | 0.54 |
| **Produce short-cut solutions in counterfactual tasks** | | | |
| Base-8 Addition (induction init.) | 0 | Add the two numbers given as input to get the output. | 0.0 |
| | 3 | Add the two numbers provided in the input. Then, adjust this sum based on the following rule: if both numbers are less than 50, add 2 to the sum. If either number is 50 or greater, add 22 to the sum. The final result is the output. | 0.35 |

Table 5: Limitations and failure cases of PE2.

| Task | Meta-prompt and Reasoning Snippets |
|---|---|
| **Neglecting instructions in the meta-prompt** | |
| Base-9 Addition | **Meta-prompt:** ... Note that the ground-truth labels are ⎵absolutely correct⎵, but the prompts (task descriptions) may be incorrect and need modification. ... |
| | **Reasoning:** No, it is not necessary to edit the prompt. The prompt is correct, but the label is incorrect. ... The issue seems to be with the label, not the prompt. |
| **Hallucination (when hints are provided in the meta-prompt)** | |
| Base-8 Addition | **Hint:** The calculation may be performed in a different numerical base. |
| | **Reasoning:** ... Given this, it's possible that the numbers are being added in base 80, not base 10. In base 80, adding 20 to the sum of two numbers would be equivalent to adding 1 in base 10. |
| | **New Prompt:** The inputs are two numbers separated by a '+'. Add these two numbers together in base 80, then add 1 to give the output in base 10. |

identifies the key step of referencing information about "today". We believe these behaviors on summarizing key steps from failure examples and incorporating them into prompts demonstrate the intelligence and capabilities of PE2.

**Limitations on following the meta-prompt and hallucination.** Despite the successes made by PE2, we note several factors that's limiting its performance in Table 5. For example, while the meta-prompt explicitly states that the "ground-truth labels are absolutely correct", the prompt proposal model insists that "the prompt is correct, but the label is incorrect" and refuses to propose a new prompt. We also attempted to guide PE2 with hints (*e.g.*, "the calculation may be performed in a different numerical base"). Regrettably, this sometimes prompts the model to generate incorrect solutions (*e.g.*, base-80), and even create rationales to verify this imagined solution. Though these observations are partly due to the difficulty of counterfactual tasks, they highlight the critical need to improve instruction following abilities and address hallucination issues in LLMs.

**Initialization is import to automatic prompt engineering.** Previously, we use "Let's think step by step." as the initialization for math reasoning tasks. We further experiment with using a *misleading* prompt, an *irrelevant* prompt and *induction* initialization (inducing from a few examples). The results are presented in Table 6 and the optimization dynamics are visualized in Fig. 3(b).

In general, performance drops when alternative initialization methods are used, which highlights the role of high-quality initialization. Still, PE2 is able to override the irrelevant or misleading prompts and gradually improve the performance (see Fig. 3(b)). Remarkably, PE2 is able to discover a high quality prompt by itself using induction initialization (84% on MultiArith-Dev) that almost matches with "Let's think step by step" (85%) designed by highly-experienced hu-

Table 6: Effect of Initialization. [†] The prompts are originally from Kojima et al. (2022).

| Initialization | MultiArith Dev | GSM8K Dev |
|---|---|---|
| default (Let's think step by step.) | 92.0 | 68.0 |
| misleading[†] (Don't think. Just feel.) | 81.0 | 50.0 |
| irrelevant[†] (It's a beautiful day.) | 73.0 | 49.0 |
| induction from few-shot examples | 84.0 | 43.0 |
| no-op (Let's think step by step.) | 85.0 | 57.0 |

man prompt engineers. This demonstrates the impressive prompt engineering capability of PE2 and suggests its potential for finding even better prompts when given additional computational resources.

## 6 RELATED WORK

**Automatic Prompt Engineering.** To alleviate the intensive trial-and-error efforts in manual prompt engineering, the research community has developed various strategies to automate this process with techniques such as incremental editing (Prasad et al., 2023), reinforcement learning (Deng et al., 2022; Zhang et al., 2022), algorithmic search (Xu et al., 2022), generating in-context demonstrations adaptively (Wan et al., 2023), among others. A notable line of work focus on leveraging LLMs themselves for automatic prompt engineering (Honovich et al., 2023; Zhou et al., 2023b; Pryzant et al., 2023). In our work, we identify potential shortcomings in these methods, subsequently introducing and rigorously examining various meta-prompt components. Our resulting method PE2 demonstrates superior performance compared to its predecessors.

**Prompting LLMs for Complex Reasoning Tasks.** Recent research works suggest that LLMs can perform complex reasoning tasks, *e.g.*, grade-school math problems (Cobbe et al., 2021). There are two major techniques to boost LLMs' performance on this: **(1) prompting methods** that guide the model to produce intermediate reasoning steps, either with few-shot demonstrations (Nye et al., 2021; Wei et al., 2022; Yao et al., 2023) or with zero-shot prompts (Kojima et al., 2022); **(2) self-reflection methods** that progressively guide the model to inspect its current output and refine it (Chen et al., 2023; Madaan et al., 2023; Paul et al., 2023; Kim et al., 2023). At its core, prompt engineering is a complex language task. Human prompt engineers usually examine the failure cases produced by the current prompt, reason and make hypotheses, and compose a new prompt. In this work, we explore these prompting strategies in building an automatic prompt engineer.

**Self-training and Self-improving for LLMs.** Self-training refers to the technique of using a weak model to annotate input-label pairs and using these pairs to train themselves (Rosenberg et al., 2005). In the context of LLMs, STaR (Zelikman et al., 2022) and Self-Improve (Huang et al., 2022) show that employing LLMs to generate high-quality reasoning chains, followed by model fine-tuning on these chains, can significantly improve the model's reasoning capabilities. In this work, we consider textual prompts as the "parameters" of LLMs, and we optimize these "parameters" with LLMs. This may be categorized as a case of self-improving (Goodman, 2023), and aligns with the motivations in recent studies (Fernando et al., 2023; Zelikman et al., 2023; Yang et al., 2023).

## 7 CONCLUSION

In this paper, we proposed and identified key components in the meta-prompt that leads to improved performance on automatic prompt engineering. The resulting method, named PE2, not only refines prompts written by human experts, but also surpasses established automatic prompt engineering baselines. Moreover, we showcased PE2's versatility by applying it to diverse language tasks, notably to counterfactual tasks, real-world applications and lengthy production prompts.

Prompt engineering a prompt engineer remains an ongoing challenge. As highlighted in our case study, we believe improving the LLM's instruction following abilities and mitigating hallucination issues will be crucial for improving automatic prompt engineering. Looking ahead, we are also excited about applying PE2 to optimize its own meta-prompt in a self-referential way, in the spirit of Metz et al. (2020); Fernando et al. (2023); Zelikman et al. (2023).

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

# A EXTENDED DISCUSSION

## A.1 DISCUSSION ON OPRO (YANG ET AL., 2023)

A recent work (Yang et al., 2023) introduced the concept of large language models as optimizers and proposed optimization by prompting (OPRO). In the following, we discuss the differences and connections between OPRO and our work.

**(1) Focus of the work.** Both OPRO and our work conduct experiments on prompt optimization; the focus of the two works differ. OPRO can be applied to general optimization problems, including linear regression and traveling salesman problem. In our work we limit the scope to prompt optimization, with a specific focus on different components of the meta-prompt.

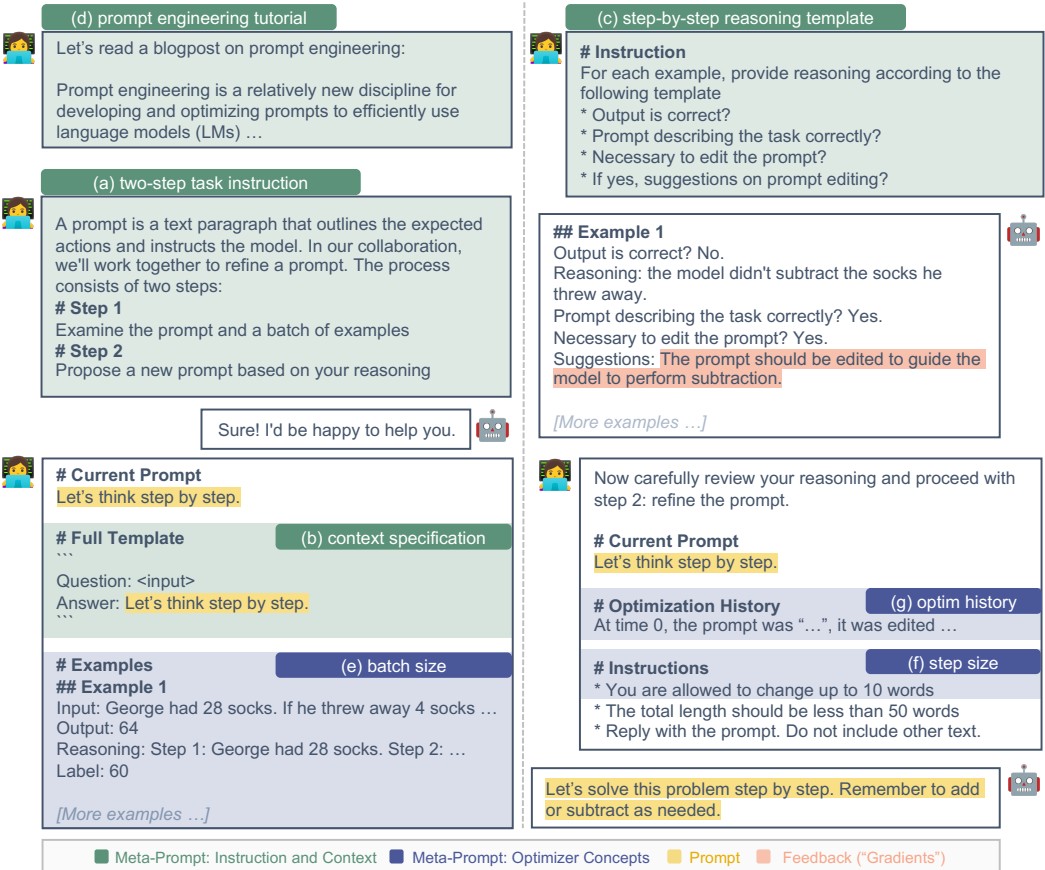

Figure 5: Illustration of meta-prompt components considered in Appendix A.2.

**(2) Optimization strategy.** The optimization strategies of the two works are different. PE2 is largely inspired by the concepts in APO (Pryzant et al., 2023), instructing the model to produce textual feedback ("gradient") explicitly. It is more analogous to gradient descent. OPRO uses the execution accuracy as rewards to guide the optimization indirectly, which, in our understanding, is more analogous to in-context RL methods (Shinn et al., 2023). For future work, it would be interesting to compare the effectiveness and efficiency of both methods in a controlled setup.

**(3) Challenges in making direct comparison.** Yang et al. (2023) mainly uses PaLM 2-L model and text-bison model as the task model (scorer), and optimizes the prompt for up to 200 steps. In our work, we mainly use text-davinci-003 and GPT-4, and optimize the prompt for 3 steps by default. Due to access and budget constraints, we are unable to make direct comparison with OPRO.

**(4) Transferability of prompts across models.** Yang et al. (2023) found a prompt, "Take a deep breath and work on this problem step-by-step", that greatly improves performance when using PaLM 2-L as the task model. We applied this prompt to text-davinci-003 and we do not observe consistent performance gains. It achieves 84.3% on MultiArith-Test and 63.2% on GSM8K-Test, which is -1.7% and +2.3% compared to "let's think step by step."

## A.2 OTHER META-PROMPT COMPONENTS WE TRIED

In addition to the meta-prompt components studies in the main paper, we also tried other components. As the results are mixed and inconclusive on these components, we report them here in the Appendix.

**Providing Detailed Instructions and Context.**

Table 7: Investigation on meta-prompt components and configurations (Appendix).

| Method | MultiArith Dev | GSM8K Dev |
|---|---|---|
| PE2 (default) | 92.0 | 68.0 |
| Meta-prompt Components | | |
| + prompt engineering tutorial | 90.0 | 63.0 |
| + tune batch size $\{1, 2, 4, 8\}$ | 92.0 | 68.0 |
| + tune step size $\{5, 10, 15, \text{None}\}$ | 95.0 | 68.0 |
| + optim history and momentum | 93.0 | 67.0 |

(d) **Prompt Engineering Tutorial.** To help the LLM better understand the task of prompt engineering, we provide an online tutorial of prompt engineering in the meta-prompt.[4]

**Incorporating Common Optimizer Concepts.** The prompt engineering problem described in Eq. 1 is essentially an optimization problem, and the prompt proposal in Eq. 2 can be considered as doing one optimization step. Thus, we consider the following concepts commonly used in gradient-based optimization and develop their verbalized counterparts to be used in our meta-prompt.

(e) **Batch Size.** Batch size is the number of (failure) examples that is used in each prompt proposal step (Eq. 2). By default PE2 uses a batch size of 2. We experiment with batch sizes of $\{1, 4, 8\}$ additionally in this section.

(f) **Step Size.** In gradient-based optimization, the step size determines the extent to which the model's weights are updated. In prompt engineering, the counterpart would be the number of words (tokens) that can be modified. We directly specify that "You are allowed to change up to $s$ words in the original prompt", where $s \in \{5, 10, 15, \text{None}\}$.[5]

(g) **Optimization History and Momentum.** Momentum (Qian, 1999) is a technique to accelerate optimization and avoid oscillations by maintaining the moving average of past gradients. To develop the verbalized counterpart of momentum, we include all past prompts (at timestamp $0, 1, ..., t - 1$), their performance on the dev set, and a summary of prompt edits.

**Results and disucssion.** We do not observe significant improvement by incorporating prompt engineering tutorial. As the tutorial is excessively long (2500+ tokens) and slows down the runtime, we do not include it in the final version of PE2. The optimizer-inspired concepts can improve the performance occasionally, but the current experiments do not give a definitive conclusion regarding their utilities. Similar to the case of gradient-based optimization, hyperparameter selection is a noisy process and tend to be task-dependant. For discrete prompt optimization, this process may be further complicated by factors such as the task model's sensitivity to prompts and the proposal model's capability to follow instructions in the meta-prompt. Additionally, momentum requires multiple optimization steps to accumulate, yet our experiments are restricted to a few optimization steps due to cost constraints.

A.3    ADDITIONAL FIGURES AND TABLES

We report the results on Date Understanding and Movie Recommendation in Table 8. We report the results on each subtask in Instruction Induction in Fig. 6. For counterfactual tasks, results using induction initialization is in Fig. 7 and results using manual initialization is in Fig. 8. Additional examples on notable prompt edits made by PE2 are in Table 9.

---

[4]https://www.promptingguide.ai/introduction. Published under MIT license.

[5]Chen et al. (2022) and Zhou et al. (2023a) showed that LLMs could follow text generation constraints specified in natural language. However Sun et al. (2023) pointed out that LLMs struggled at meeting fine-grained constraints.

| Method | Final Prompt | Test Acc. |
|--------|-------------|-----------|
| Date Understanding | | |
| Zero-shot CoT | Let's think step by step. | 0.391 |
| Iterative APE | Let's dissect it and ponder over each phase. | 0.467 |
| APO | Determine the exact date from the scenario, considering cultural date formats, time zones, and periods. Use the provided date as a reference. Account for leading zeros, leap years, relative dates, and event-based time references. Provide the result in MM/DD/YYYY format. | 0.450 |
| PE2 | Analyzing the given information, let's calculate the solution. Remember to consider the context provided, such as references to 'today' or specific dates. | 0.544 |
| Movie Recommendation | | |
| Zero-shot CoT | Let's think step by step. | 0.570 |
| Iterative APE | Let's dissect it and consider every step in order. | 0.673 |
| APO | Identify the movie that shares the most significant themes and narrative structure with the given movies. Prioritize these factors over tone and pacing. Choose the most similar movie from the options, explaining your choice. | 0.750 |
| PE2 | Considering factors such as genre, director, actors, release period, audience target, animation style, and humor, analyze the similarities among the given movies and identify the movie from the options that shares the most similarities. | 0.790 |

Table 8: Results on Date Understanding and Movie Recommendation from BIG-bench Hard (Suzgun et al., 2023). BIG-bench Hard is derived from BIG-bench (BIG-bench Authors, 2023). In these experiments, we use gpt-3.5-turbo-instruct as the task model and gpt-4 as the prompt proposal model.

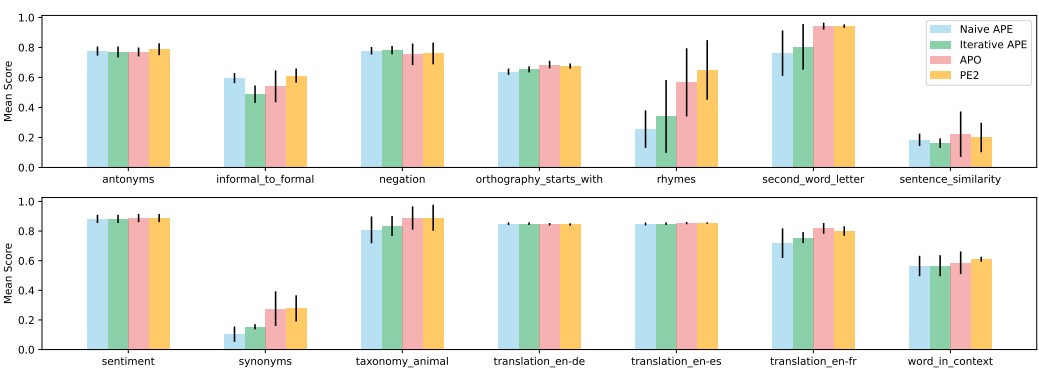

Figure 6: Results on the Instruction Induction Benchmark. The performance of APO and PE2 are close to each other on most tasks. Our hypothesis is that tasks in Instruction Induction Benchmark are relatively easier compared to the other benchmarks, leading to performance saturation.

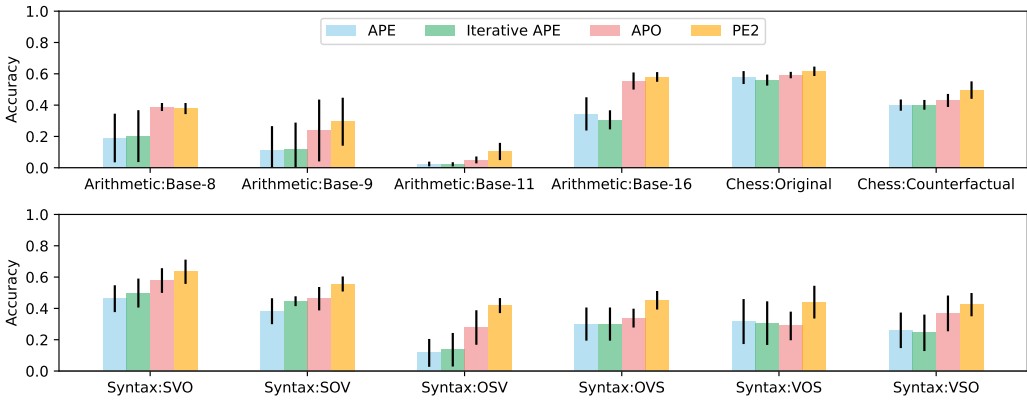

Figure 7: Results on counterfactual tasks (induction initialization).

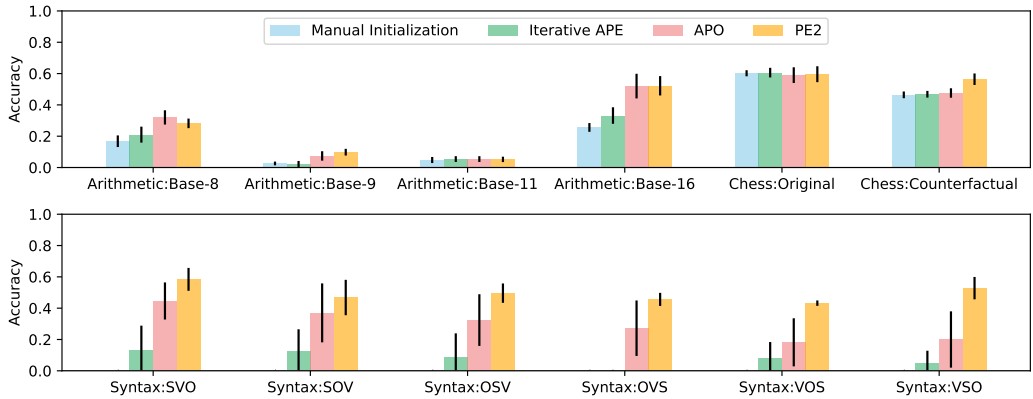

Figure 8: Results on counterfactual tasks (manual initialization).

Table 9: Notable prompt edits made by PE2 (Part 2).

| Task | $t$ | Prompt | Dev Acc. |
|------|-----|--------|----------|
| **Correct wrong or incomplete task instructions** | | | |
| Antonyms | 0 | Write the opposite of the given word by adding an appropriate prefix. | 0.3 |
| | 1 | Find the opposite of the given word. If applicable, add or remove an appropriate prefix to form the opposite. | 0.6 |
| **Provide more specific context and details** | | | |
| Sentence Similarity | 0 | Rate the similarity between Sentence 1 and Sentence 2 on a scale from 1 to 5, with 1 being 'probably not similar' and 5 being 'perfectly similar'. | 0.0 |
| | 1 | Rate the similarity between Sentence 1 and Sentence 2 as '1 - probably not similar', '2 - possibly', '3 - moderately', '4 - almost perfectly', or '5 - perfectly similar'. | 0.15 |
| **Produce short-cut solutions in counterfactual tasks** | | | |
| Base-9 Addition (induction init.) | 0 | Add the numbers in each input together to get the output. | 0.0 |
| | 1 | Add the numbers in each input together and then add 11 to get the output. | 0.2 |

## B  META-PROMPTS

We implement the meta-prompts using the guidance toolkit[6], which enables multi-round conversations and supports basic handlebars-style syntax to control the workflow.

### B.1  INITIALIZATION PROMPT $p^{init}$

The initialization prompt is originally from APE (Zhou et al., 2023b). In this paper, it is shared by all methods (Iterative APE, APO and PE2).

```
1   {{#system~}}
2   You are a helpful assistant.
3   {{~/system}}
4
5   {{#user~}}
6   I gave a friend an instruction and {{n_demo}} inputs. The friend read the instruction and
        wrote an output for every one of the inputs.
7   Here are the input-output pairs:
8
9   {{demos}}
10
11  What was the instruction? It has to be less than {{max_tokens}} tokens.
12  {{~/user}}
13
14  {{#assistant~}}
15  The instruction was {{gen 'instruction' [[GENERATION_CONFIG]]}}
16  {{~/assistant}}
```

---

[6]https://github.com/guidance-ai/guidance

## B.2 APE

```
1  {{#system~}}
2  You are a helpful assistant.
3  {{~/system}}
4
5  {{#user~}}
6  Generate a variation of the following instruction while keeping the semantic meaning.
7
8  {{prompt}}
9
10 The new instruction has to be less than {{max_tokens}} words.
11 Reply with the new instruction. Do not include other text.
12 {{~/user}}
13
14 {{#assistant~}}
15 {{gen 'new_prompt' [[GENERATION_CONFIG]]}}
16 {{~/assistant}}
```

## B.3 APO

### Part 1 - Generating "gradients"

```
1  {{#system~}}
2  You are a helpful assistant.
3  {{/system~}}
4
5  {{#user~}}
6  I'm trying to write a zero-shot classifier prompt.
7
8  My current prompt is:
9  "{{prompt}}"
10
11 But this prompt gets the following examples wrong:
12 {{failure_string}}
13
14 Give {{n_reasons}} reasons why the prompt could have gotten these examples wrong. Do not
         include other text.
15 {{/user~}}
16
17 {{#assistant~}}
18 {{gen 'gradients' temperature=0.0}}
19 {{/assistant~}}
```

### Part 2 - Refining the prompt

```
1  {{#system~}}
2  You are a helpful assistant.
3  {{/system~}}
4
5  {{#user~}}
6  I'm trying to write a zero-shot classifier.
7
8  My current prompt is:
9  "{{prompt}}"
10
11 But it gets the following examples wrong:
12 {{failure_string}}
13
14 Based on these examples the problem with this prompt is that:
15 {{gradient}}
16
17 Based on the above information, I wrote an improved prompt. The total length of the prompt
        should be less than {{max_tokens}} words.
18 {{/user~}}
19
20 {{#assistant~}}
21 The improved prompt is {{gen 'new_prompt' temperature=0.0}}
22 {{/assistant~}}
```

## B.4 PE2

```
1  {{#system~}}
2  You are a helpful assistant.
```

```
3   {{~/system}}
4
5   {{#if instruction}}
6   {{#user~}}
7   Let's read a blogpost on prompt engineering:
8   {{instruction}}
9   {{~/user}}
10  {{/if}}
11
12  {{#user~}}
13  A prompt is a text paragraph that outlines the expected actions and instructs the model to
        generate a specific output. This prompt is concatenated with the input text, and the
        model then creates the required output.
14
15  In our collaboration, we'll work together to refine a prompt. The process consists of two main
        steps:
16
17  ## Step 1
18  I will provide you with the current prompt, how the prompt is concatenated with the input text
        (i.e., "full template"), along with {{batch_size}} example(s) that are associated with
        this prompt. Each examples contains the input, the reasoning process generated by the
        model when the prompt is attached, the final answer produced by the model, and the ground
        -truth label to the input. Your task is to analyze the examples, determining whether the
        existing prompt is decsribing the task reflected by these examples precisely, and suggest
        changes to the prompt.
19
20  ## Step 2
21  Next, you will carefully review your reasoning in step 1, integrate the insights to craft a
        new, optimized prompt. Optionally, the history of refinements made to this prompt from
        past sessions will be included. Some extra instructions (e.g., the number of words you
        can edit) will be provided too.
22  {{~/user}}
23
24  {{#assistant~}}
25  Sure, I'd be happy to help you with this prompt engineering problem.
26  Please provide me with the prompt engineering history, the current prompt, and the examples
        you have.
27  {{~/assistant}}
28
29  {{#user~}}
30  ## Prompt
31  {{prompt}}
32
33  ## Full Template
34  This describes how the prompt of interested is concatenated with the input text.
35  The prompt may appear before the input text, or after the input text.
36  Optionally the full template may contain other template information.
37  ```
38  {{full_prompt}}
39  ```
40
41  ## Examples
42  {{examples}}
43
44  ## Instructions
45  For some of these examples, the output does not match with the label. This may be due to the
        prompt being misleading or not describing the task precisely.
46
47  Please examine the example(s) carefully. Note that the ground-truth labels are __absolutely
        correct__, but the prompts (task descriptions) may be incorrect and need modification.
        For each example, provide reasoning according to the following template:
48
49  ### Example <id>
50  Input: <input>
51  Output: <output>
52  Label: <label>
53  Is the output correct compared to the label: <yes or no, and your reasoning>
54  Is the output correctly following the given prompt: <yes or no, and your reasoning>
55  Is the prompt correctly describing the task shown by the input-label pair: <yes or no, and
        your reasoning>
56  To output the correct label, is it necessary to edit the prompt: <yes or no, and your
        reasoning>
57  If yes, provide detailed analysis and actionable suggestions to edit the prompt: <analysis and
        suggestions>
58  {{~/user}}
59
60  {{#assistant~}}
61  {{gen 'reasoning' temperature=0}}
62  {{~/assistant}}
63
64  {{#user~}}
```

```
65  Now please carefully review your reasoning in Step 1 and help with Step 2: refining the prompt
        .
66
67  {{#if history}}
68  ## Prompt Refinement History from the Past
69  Note that higher accuracy means better. If some edits are useful in the past, it may be a good
        idea to make edits along the same direction.
70  {{history}}
71  {{/if}}
72
73  ## Current Prompt
74  {{prompt}}
75
76  ## Instructions
77  {{#if step_size}}
78  * You are allowed to change up to {{step_size}} words in the original prompt.
79  {{/if}}
80  {{#if max_tokens}}
81  * The total length of the prompt should be less than {{max_tokens}} words.
82  {{/if}}
83  * Please help edit the prompt so that the updated prompt will not fail on these examples
        anymore.
84  * Reply with the prompt. Do not include other text.
85  {{~/user}}
86
87  {{#assistant~}}
88  {{gen 'new_prompt' temperature=0.7 max_tokens=300}}
89  {{~/assistant}}
90
91  {{#if history}}
92  {{#user~}}
93  Now please summarize what changes you've made to the prompt, in the following format. Make
        sure the summariy is concise and contains no more than 200 words.
94
95  " * At step {{timestamp}}, the prompt has limitations such as <summary of limitations>.
        Changes to the prompt include <summary of changes>."
96
97  Reply with the summarization. Do not include other text.
98  {{~/user}}
99
100 {{#assistant~}}
101 {{gen 'new_history' temperature=0.7 max_tokens=200}}
102 {{~/assistant}}
103 {{/if}}
```

## C  LONG PROMPT OPTIMIZATION

We conduct additional experiments on the long production prompt, to better understand the sensitivity of PE2 to hyper-parameters for *long* prompts. Note that in the other three tasks (math reasoning, instruction induction, counterfactual eval) the prompts are typically shorter than 50 tokens, while the production prompt has 5k+ tokens, which calls for a separate investigation.

### C.1  ABLATION FOR LONG PROMPT OPTIMIZATION

We looked at the impact on the final test score when modifying the set of optimization parameters; *batch size* $\in \{3, 10\}$ and *step size* $\in \{10, 40, 50\}$ given in Section 3 and the *LLM* compute $m \in \{2, 4\}$ and *search budget* $n \in \{2, 4\}$ given in Section 4.2.

**Dataset:**  We first created a smaller evaluation dataset from the full production prompt dataset, which consists of the instructions for $< 50\%$ of the Domains given in the original data, and a small uniform sub-sample ($< 20\%$) of the questions for these domains, we did not modify the k-shot examples used to the align them to the subset of Domains used.

We ran 3-fold cross-validation using the evaluation dataset with train ($40\%$), validation ($30\%$) and testing ($30\%$). Figure 9 reports the absolute change in F1-score at each time-step for a total of up to 10 steps in each experiment.

The highest performing prompt generated by PE2 improves the F1 test score by $+11$ points, and the overall trend is a positive and significant (Pearson correlation $r = 0.55, p = 1.68e^{-5}$), there is

however a significant margin between the maximum (0.10) and the mean ($0.02 \pm 0.11$) change in the score.

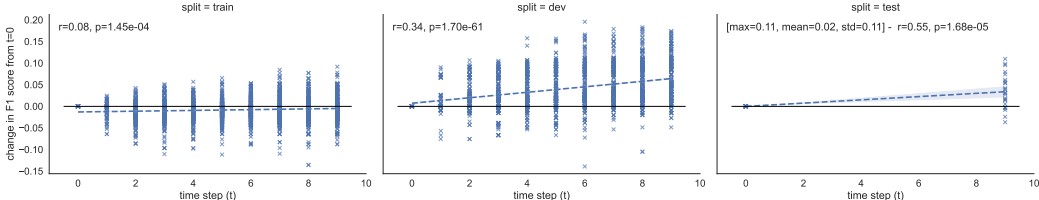

Figure 9: The distribution and trend of the performance of PE2 across the 3-fold cross validation.

### C.1.1 IMPACT OF OPTIMIZATION PARAMETERS

Figure 10 gives the ablation results for varying the *batch size* (Figure 10(a)) and *step size* (Figure 10(b)). The *batch size* controls the number of examples the LLM can inspect in order to diagnose the error and suggest a solution, while the *step size* controls the magnitude of a change the LLM can make to the text.

With *batch size* $\in \{3, 10\}$, we observe that larger batch sizes have a more significant impact on improving the performance on the test set. From our ablation, although the best performing prompts in the two cases have the same score, for *batch size*=10, the Pearson correlation is $r = 0.59, p = 2.46e^{-4}$ vs $r = 0.49, p = 2.84e^{-2}$ for *batch size*=3.

The impact of the *step size* $\in \{10, 40, 50\}$ appears to be a little more noisy with our task. The best result is from *step size*=50, but the most significant result is observed with with *step size*=40 with Pearson correlation, $r = 0.52, p = 4.2e^{-3}$, which is an order of magnitude smaller than the other two settings, *step size* $\in \{10, 50\}$, where *step size*=50 achieves a better score.

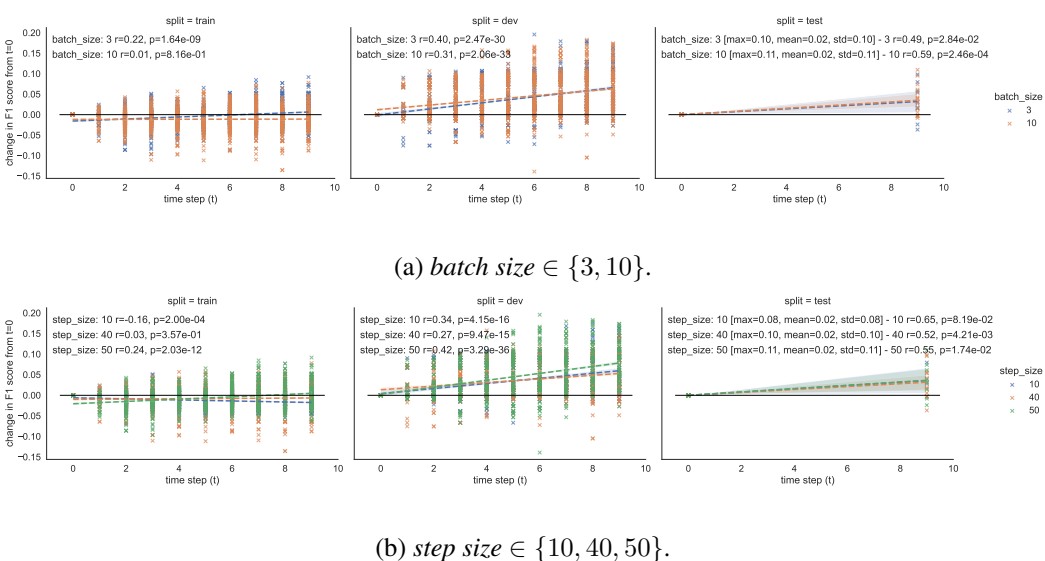

(a) *batch size* $\in \{3, 10\}$.

(b) *step size* $\in \{10, 40, 50\}$.

Figure 10: The impact of varying the optimization parameters *batch size* and *step size*. Figures report the Pearson coefficient ($r$) and the p-value ($p$) for each grouping. For the test split, Figures additionally report the score of the best performing prompt in the set ($max$), the mean score ($mean$) of the prompts and its standard deviation ($std$).

### C.1.2 IMPACT OF COMPUTE PARAMETERS

Figure 11 gives the ablation results for varying $n$ and $m$, two hyperparameters defined in §4.2. $n$ controls the number of candidate prompts selected for generating new prompts, while the $m$

determines the number of new prompts each candidate generates. Together these two parameters enable the optimization process to i) inspects $n$-more candidate solutions and ii) generate $n \times m$ more candidate solutions for evaluation in the next step.

For both parameters, improvement in test performance is positively correlated with increasing the value, however, increasing the *Search* budget ($n$) has a more significant p-value; $r = 0.60, p = 1.73e^{-4}$ for $n = 4$ vs $r = 0.62, p = 3.59e^{-3}$ for $m = 4$.

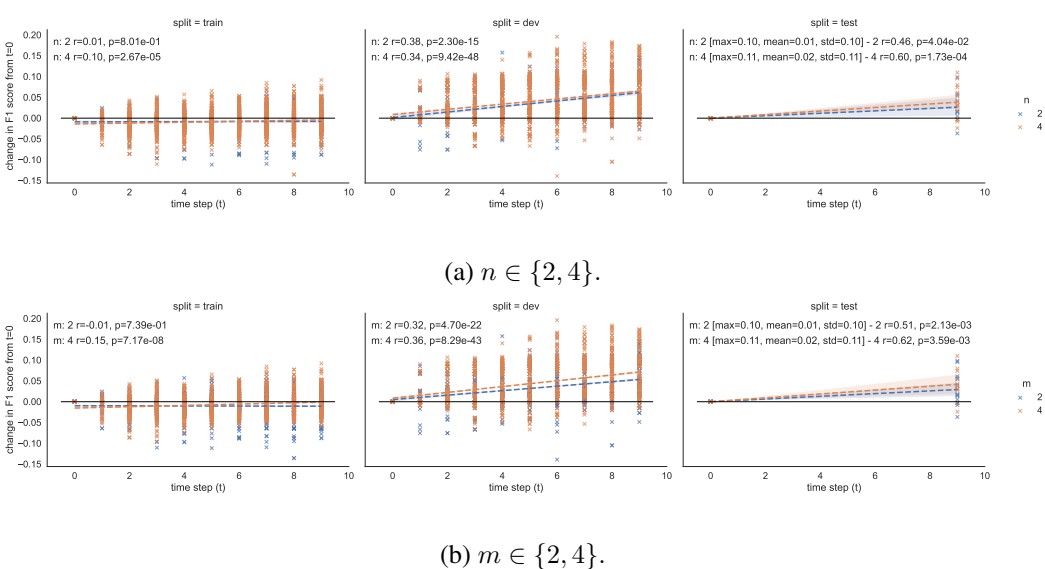

(a) $n \in \{2, 4\}$.

(b) $m \in \{2, 4\}$.

Figure 11: The impact of varying $n$ and $m$. Figures report the Pearson coefficient ($r$) and the p-value ($p$) for each grouping. For the test split, Figures additionally report the score of the best performing prompt in the set ($max$), the mean score ($mean$) of the prompts and its standard deviation ($std$).

### C.2 PERFORMANCE ON THE COMPLETE PRODUCTION PROMPT

**Experiment configuration:** We ran the a 2-fold cross validation with on the full production prompt dataset described in Section 4.1. This is the setting that we report in Fig. 1.

The data was divided between training (50%), validation (25%) and testing (25%). *step size* is 200, *batch size* is 15, with random selection of batch items. The number of candidate prompts inspected at each step is $n \in \{2, 4\}$, the number of new prompts generated per candidate $m = 2$.

We use the F1-score for evaluating model outputs and report the absolute change in score with the initial prompt after optimizing for up to 10 time steps.

Figure 12(a) gives the results for our full long-prompt optimization. The overall test performance shows a positive correlation the score difference; Pearson correlation $r = 0.47, p = 2.39e^{-1}$, but the trend is not significant. Figure 12(b) separates the results between $n \in \{2, 4\}$, and see that for $n = 4$ the Pearson correlation is positive ($r = 0.82, p = 1.82e^{-1}$), while for $n = 2$ the correlation is negative ($r = -0.22, p = 7.77e^{-1}$).

## D REPRODUCIBILITY

### D.1 TASKS AND DATA

We summarize the dataset size and data split information in Table 10. We summarize the source and license information of the datasets in Table 11.

**(1) Mathematical Reasoning.** The MultiArith dataset (Roy & Roth, 2015) contains 600 examples. As our prompt optimization method requires a training set, we randomly split into 100/100/400 for

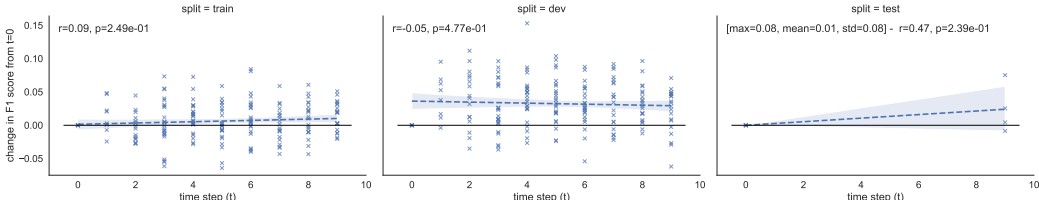

(a) The over all performance of PE2 on the full production prompt and dataset. 2-fold cross-validation was used to obtain results with $n \in 2, 4$

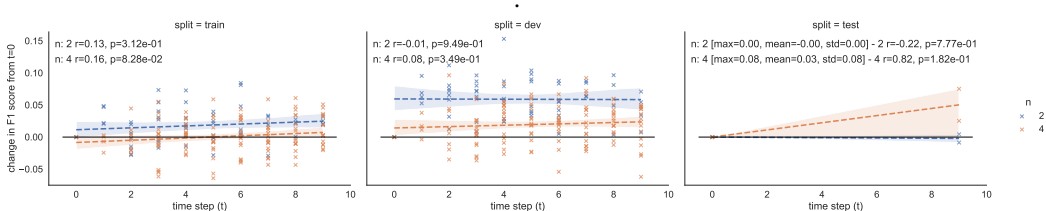

(b) The impact of the *Search* budget ($n$). We see that increasing the candidate solution space PE2 can expand from $n = 2$ to $n = 4$ strongly changes the direction of test the correlation and significantly improves the result.

Figure 12: Large Prompt Optimization with PE2. Figures report the Pearson coefficient ($r$) and the p-value ($p$) for each grouping. For the test split, Figures additionally report the score of the best performing prompt in the set ($max$), the mean score ($mean$) of the prompts and its standard deviation ($std$).

train/dev/test. This creates a slight discrepancy when comparing the results with past reported results. We ensure our reproduction is fair across different methods by using this fixed split. The GSM8K dataset (Cobbe et al., 2021) has a provided test split (1319 examples). We randomly selected 200 examples for the original train split, and use 100 as $D_{train}$ and 100 as $D_{dev}$.

**(2) Instruction Induction.** We closely follow the settings in (Zhou et al., 2023b). For each sub-task, we randomly sample 5 different $D_{train}/D_{dev}/D_{test}$ of size 100/20/100. We list the sub-tasks in Instruction Induction benchmark in Table 12. We removed 8 tasks (active to passive, diff, first word letter, letters list, num to verbal, singular to plural, sum), because our baseline method APE Zhou et al. (2023b) already achieves near perfect accuracies (95%+) on these tasks. We also removed 2 tasks (cause and effect, common concept) because they have less than 50 examples in total, and it is challenging to create train/dev/test split from these examples.

**(3) Counterfactual Evaluation.** We use three subtasks in this evaluation suite: arithmetic, chess and syntax. For each subtask, we randomly sample 5 different $D_{train}/D_{dev}/D_{test}$ of size 100/20/100.

**(4) Production Prompt.** We use a randomly sampled sub-set of human annotated queries and labels ($> 150$), which are derived from user reported errors. The data is divided between training (50%), validation (25%) and testing (25%). We use the F1-score for evaluating model outputs and report the absolute change in score with the initialization prompt. More details in §C.

**(5) BIG-bench Hard Tasks.** We use two tasks, Date Understanding and Movie Recommendation, that are close to real-world applications and highlighted in the OPRO paper (Yang et al., 2023). We randomly sample 100/100/500 examples for $D_{train}/D_{dev}/D_{test}$.

| Dataset | Subtasks | $|T_{train}|$ | $|T_{dev}|$ | $|T_{test}|$ | # Random Samples |
|---|---|---|---|---|---|
| MultiArith (Roy & Roth, 2015) | - | 100 | 100 | 400 | 1 |
| GSM8K (Cobbe et al., 2021) | - | 100 | 100 | 1319 | 1 |
| Instruction Induction (Honovich et al., 2023) | 14 Subtasks | 100 | 20 | 100 | 5 |
| Counterfactual Eval (Wu et al., 2023) | 12 Subtasks | 100 | 20 | 100 | 5 |
| BIG-Bench Hard (Suzgun et al., 2023) | 2 Subtasks | 100 | 100 | 500 | 1 |

Table 10: Dataset sizes and data splits.

| Dataset | License | Source |
|---|---|---|
| MultiArith (Roy & Roth, 2015) | Unknown | `https://github.com/wangxr14/Algebraic-Word-Problem-Solver/` |
| GSM8K (Cobbe et al., 2021) | MIT | `https://github.com/openai/grade-school-math` |
| Instruction Induction (Honovich et al., 2023) | Apache-2.0 | `https://github.com/orhonovich/instruction-induction` |
| Counterfactual Eval (Wu et al., 2023) | Unknown | `https://github.com/ZhaofengWu/counterfactual-evaluation` |
| BIG-bench Hard (Suzgun et al., 2023) | Apache-2.0 | `https://github.com/google/BIG-bench` |

Table 11: License and Source of the datasets used in this study.

## D.2 EXTENDED EXPERIMENT DETAILS

By default the max length of prompts are set to be 50 tokens, following Zhou et al. (2023b). For counterfactual tasks, to allow more space to explain the counterfactual situations, the max length is set to be 200 tokens.

## D.3 PROMPTS FOUND BY EACH METHOD

See Table 12-17.

| Task | Instruction | Demonstration |
|------|-------------|---------------|
| Subtasks used in this work (14) | | |
| Second Letter | Extract the first letter of the input word. | cat → a |
| Starting With | Extract the words starting with a given letter from the input sentence. | The man whose car I hit last week sued me. [m] → man, me |
| Negation | Negate the input sentence. | Time is finite → Time is not finite. |
| Antonyms | Write a word that means the opposite of the input word. | won → lost |
| Synonyms | Write a word with a similar meaning to the input word. | alleged → supposed |
| Membership | Write all the animals that appear in the given list. | cat, helicopter, cook, whale, frog, lion → frog, cat, lion, whale |
| Rhymes | Write a word that rhymes with the input word. | sing → ring |
| Informal to Formal | Rephrase the sentence in formal language. | Please call once you get there → Please call upon your arrival. |
| Translation EN-DE | Translate the word into German. | game → spiel |
| Translation EN-ES | Translate the word into Spanish. | game → jeugo |
| Translation EN-FR | Translate the word into French. | game → jeu |
| Sentiment | Determine whether a movie review is positive or negative. | The film is small in scope, yet perfectly formed. → positive |
| Sentence Similarity | Rate the semantic similarity of two sentences on a scale of 0 to 5 | Sentence 1: A man is smoking. Sentence 2: A man is skating. → 0 - definitely not |
| Word in Context | Determine whether an input word has the same meaning in the two sentences. | Sentence 1: Approach a task. Sentence 2: To approach the city. Word: approach → not the same |
| Subtasks removed due to near-perfect accuracy (95%+) with baseline method (8) | | |
| First Letter | Extract the first letter of the input word. | cat → c |
| List Letters | Break the input word into letters, separated by spaces. | cat → c a t |
| Singular to Plural | Convert the input word to its plural form. | cat → cats |
| Active to Passive | Write the input sentence in passive form. | The artist introduced the scientist. → The scientist was introduced by the artist. |
| Larger Animal | Write the larger of the two given animals. | koala, snail → koala |
| Sum | Sum the two given numbers. | 22 10 → 32 |
| Diff | Subtract the second number from the first. | 32 22 → 10 |
| Number to Word | Write the number in English words. | 26 → twenty-six |
| Subtask removed due to small dataset size (2) | | |
| Cause and Effect | Find which of the two given cause and effect sentences is the cause. | Sentence 1: The soda went flat. Sentence 2: The bottle was left open. → The bottle was left open. |
| Common Concept | Find a common characteristic for the given objects. | guitars, pendulums, neutrinos → involve oscillations. |

Table 12: Details of Instruction Induction dataset. Adapted from Table 4 in Honovich et al. (2023).

| Task | Category | Demonstration |
|------|----------|---------------|
| Arithmetic - Two-digit addition | | |
| Base-10 | Original | 22+10 → 32 |
| Base-8 | Counterfactual | 76+76 → 174 |
| Base-9 | Counterfactual | 76+14 → 101 |
| Base-11 | Counterfactual | 76+14 → 8A |
| Base-16 | Counterfactual | EC+DD → 1C9 |
| Chess - Legality of a 4-move opening | | |
| Normal Rules | Original | 1. g3 Ng6 2. b3 Kf8 * → illegal |
| Swapping bishops and knights | Counterfactual | 1. g3 Ng6 2. b3 Kf8 * → legal |
| Syntax - Identify the main subject and the main verb of a sentence | | |
| SVO | Original | he has good control . → he has |
| SOV | Counterfactual | he good control has . → he has |
| VSO | Counterfactual | has he good control . → he has |
| VOS | Counterfactual | has good control he . → he has |
| OVS | Counterfactual | good control has he . → he has |
| OSV | Counterfactual | good control he has . → he has |

Table 13: Details of Couterfactual Evaluation dataset.

Table 14: Prompts find by prompt optimization methods. For instruction induction, experiments were run with 5 random data splits; In this table we report the prompts found in one run (seed=0).

| Task | Method | Prompt |
|------|--------|--------|
| **Math Reasoning** | | |
| MultiArith | Zero-shot CoT | Let's think step by step. |
| | APE | Let's work this out in a step by step way to be sure we have the right answer. |
| | Iterative APE | Let's proceed in a methodical, step-by-step manner. |
| | APO | Given the scenario, perform the necessary calculations step by step to find the final result. Consider all parts of the input and the sequence of events. |
| | PE2 | Let's solve this problem by considering all the details. Pay attention to each piece of information, remember to add or subtract as needed, and perform the calculations step by step. |
| GSM8K | Zero-shot CoT | Let's think step by step. |
| | APE | Let's work this out in a step by step way to be sure we have the right answer. |
| | Iterative APE | Let's dissect this and tackle it gradually, one phase at a time. |
| | APO | Given the scenario, perform necessary calculations and provide a step-by-step explanation to arrive at the correct numerical answer. Consider all information provided. |
| | PE2 | Let's solve the problem step-by-step and calculate the required total value correctly. |
| **Instruction Induction** | | |
| antonyms | APO | Provide the opposite or a negative form of the given input word. |
| | PE2 | Provide the opposite or a negative form of the given input word. |
| informal_to_formal | APO | Convert each sentence into a formal version, preserving the original structure, meaning, and tone. Avoid excessive formality, unnecessary changes, and maintain idiomatic expressions. Handle contractions appropriately. |
| | PE2 | Please transform each sentence into a version that maintains the original meaning but is expressed in a more formal or polite manner. |
| negation | APO | Negate the statement given in the input. |
| | PE2 | Negate the statement given in the input. |
| orthography_starts_with | APO | Identify the word or phrase in the sentence that starts with the given letter, considering the context and grammar. Include articles if they precede the word or phrase. |
| | PE2 | Find the word or phrase in the sentence that starts with the given letter, and write it as the output. |
| rhymes | APO | Remove the first letter of the given word. Find a word that rhymes with the remaining part, has the same syllable count, and is not a derivative or the same as the original word. |
| | PE2 | Generate a word that rhymes with the given word. |
| second_word_letter | APO | Identify the second character from the start in each input word and provide it as the output. |
| | PE2 | Identify the second character from the start of the given word. |
| sentence_similarity | APO | Rate the similarity between Sentence 1 and Sentence 2 using the scale: 1 - 'probably not', 2 - 'possibly', 3 - 'probably', 4 - 'likely', 5 - 'perfectly'. |
| | PE2 | Rate the similarity between Sentence 1 and Sentence 2 using the scale: 1 - 'probably not', 2 - 'possibly', 3 - 'probably', 4 - 'likely', 5 - 'perfectly'. |
| sentiment | APO | Determine if the given movie review statement is positive or negative. |
| | PE2 | Determine if the given movie review statement is positive or negative. |
| synonyms | APO | Provide a single word that is closely related to the given input, considering its most common usage. |
| | PE2 | Identify a word that is closely connected, in meaning or context, with the provided input word. |
| taxonomy_animal | APO | Remove all items from the list that are not animals. |
| | PE2 | Remove all items from the list that are not animals. |
| translation_en-de | APO | Translate each English word into German. |
| | PE2 | Translate each English word into German. |
| translation_en-es | APO | Provide the most commonly used Spanish translation for the given English word. |
| | PE2 | Translate the given term from English to Spanish. Note that the translation may be a single word or a phrase. |
| translation_en-fr | APO | Provide the French equivalent for the given English word. |
| | PE2 | Translate the following word from English to its most common equivalent in French. |
| word_in_context | APO | Determine if the word provided is used in the same sense/context in both sentences. If it is, write 'same.' If not, write 'not the same.' |
| | PE2 | Determine if the word provided is used in the same sense/context in both sentences. If it is, write 'same.' If not, write 'not the same.' |

Table 15: Prompts find by prompt optimization methods. Experiments were run with 5 random data splits; In this table we report the prompts found in one run (seed=0).

| Task | Method | Prompt |
|---|---|---|
| Counterfactual Evaluation (Induction Initialization) | | |
| arithmetic_base11 | APO | Given two numbers in hexadecimal format (0-9, A-F), convert each number to decimal. Add the two decimal numbers together. Output the sum in hexadecimal format. If the sum exceeds the range of a single hexadecimal digit (0-F), represent it appropriately in hexadecimal. For example, if the input is 'A' and 'B', the output should be '15' as 'A' is 10 and 'B' is 11 in decimal, and their sum is 21 which is '15' in hexadecimal. |
| | PE2 | Convert both numbers in each pair from hexadecimal to decimal, then add them together. Output the resultant sum in hexadecimal. For instance, if the input is A4+61, convert A4 and 61 to decimal (164 and 97 respectively), add them together to get 261, and convert this back to hexadecimal to get 105. |
| arithmetic_base16 | APO | Given two hexadecimal numbers as input, add them together using base 16 arithmetic. The input hexadecimal numbers will be in uppercase and may have different number of digits. Align the numbers from right to left, similar to traditional addition, and handle any overflow or carry appropriately. Output the sum as an uppercase hexadecimal number. |
| | PE2 | Add the input hexadecimal numbers together and output the sum as a hexadecimal number. For example, if the input is "44+E7", the output should be "12B", because the sum of hexadecimals 44 and E7 equals 12B in hexadecimal. |
| arithmetic_base8 | APO | Given an input string containing two numbers separated by a '+', calculate the sum of these two numbers. Then, add 20 to this sum to get the output. For example, if the input is '22+47', first add 22 and 47 to get 69, then add 20 to 69 to get the final output of 89. Similarly, if the input is '74+26', first add 74 and 26 to get 100, then add 20 to 100 to get the final output of 120. The '+' symbol should be interpreted as an addition operator, and the order of operations should be to add the two numbers first, then add 20 to the sum. The input will always be formatted correctly, with no spaces or other characters around the '+' symbol. |
| | PE2 | To find the correct output, first add the two numbers given as input. Once you have the sum of these two numbers, add an additional 22 to this sum. For example, if the input is "17+65", you should first add 17 and 65 to get 82, then add 22 to 82. The correct output in this case would be 104. |
| arithmetic_base9 | APO | Add the numbers together. |
| | PE2 | Add the two numbers given as input and then add 10 to the result to generate the output. For example, if the input is '25+18', the output should be '53' because 25 plus 18 equals 43, and adding 10 gives 53. |
| chess_cf | APO | Determine if the given sequence of chess moves, starting from the initial game position, is legal or not according to the standard rules of chess. Consider the unique movements and restrictions of each piece, the alternating turns of the players (white and black), and the entire game state up to the given point. Evaluate the sequence as a whole, not just individual moves. Note that the sequence ends with an asterisk (*). |
| | PE2 | Please assess the legality of the following sequence of chess moves based on standard chess rules. If all moves are valid according to the rules of chess, indicate "Legal." If there is any move that violates standard chess rules, respond with "Illegal". For example, if the sequence is "1. e4 e5 2. Nf3 d6", your response should be "Legal". If the sequence is "1. e4 e5 2. Kf2", your response should be "Illegal" because the king cannot be exposed to check. |
| chess_original | APO | Determine if the given sequence of chess moves is legal or illegal. |
| | PE2 | Determine if the given sequence of chess moves is legal or illegal. |

Table 16: Prompts find by prompt optimization methods. Experiments were run with 5 random data splits; In this table we report the prompts found in one run (seed=0).

| Task | Method | Prompt |
|---|---|---|
| Counterfactual Evaluation (Induction Initialization) - Continued | | |
| syntax_osv | APO | Identify the main subject and verb in the sentence. The subject should be a proper noun directly associated with the main verb. Focus on the main clause that conveys the primary information. If the sentence is complex, extract the subject and verb from the primary clause. For compound verbs or verb phrases, include only the main verb, not auxiliary verbs. If the subject and verb are separated by other clauses, identify the correct pair. If the subject is implied, make a reasonable guess. Write the subject and verb as a pair in the output. |
| syntax_osv | PE2 | Identify the subject and verb at the end of the sentence. The subject may not always be a proper noun. The verb should be in the present tense. Write them out as a pair in the output. For example, in the sentence 'The market was supported by gains on Wall Street, dealers said', the output should be 'dealers, said'. |
| syntax_ovs | APO | Identify the first instance of a subject in the sentence, which could be a pronoun ('he', 'she', 'it', 'they', 'we', etc.) or a noun/noun phrase. Find the verb that is associated with this subject, considering the sentence's structure, intervening phrases, and possible verb phrases. The verb may not directly follow the subject and could precede it. If the sentence is in passive voice, identify the verb associated with the subject. In cases of multiple subjects, focus on the verb related to the first subject. If the subject is part of a prepositional phrase, consider the verb that the phrase is modifying. Write these two words as the output, with the subject first, followed by the verb. |
| syntax_ovs | PE2 | Identify the first personal pronoun in the sentence and find the verb that is semantically linked to it. Write these two words as your output. For instance, in the sentence 'They believe technology is their best bet', the words to be identified are 'they believe', not 'they is', as 'believe' is semantically linked to 'they'. |
| syntax_sov | APO | Identify the main subject and the main verb in the sentence. Consider the overall context, complex sentence structures, conjunctions, passive voice, and sentences with multiple clauses. Output the main subject and the main verb together, as they appear in the input. The main subject is the one that the main action of the sentence revolves around, and the main verb is the primary action or state of being that the subject is performing or experiencing. |
| syntax_sov | PE2 | Identify the subject and the main verb in the sentence and write them together in the same order as they appear in the sentence, excluding any additional words in between. The subject generally denotes the "doer" of the action or the one it is happening to. The main verb expresses the action or state of being. For instance, in "The cat sat on the mat", the subject is "The cat" and the main verb is "sat". So, the output should be "The cat sat". Ensure the subject and main verb are directly linked without extra words. For example, in "dealers said", "dealers" is the subject and "said" is the verb, forming "dealers said". |
| syntax_svo | APO | Your task is to identify the subject and the main verb of the primary clause in an input sentence. Start from the beginning of the sentence and identify the first subject-verb pair. Ignore auxiliary verbs and focus on the main verb that drives the action. If the sentence has multiple clauses, focus on the first one that forms a complete thought. Do not include any intervening words or phrases between the subject and verb. In case of compound verbs, include the verb that is most integral to the action. Ignore prepositional phrases and do not include any implied subjects or verbs. Your output should be concise, containing only the subject and the main verb. |
| syntax_svo | PE2 | Read the input sentence and identify the subject and the verb of the main clause. Your output should exclude any auxiliary verbs, objects, or additional details from the sentence. For example, if the input is "John is eating an apple", the output should be "John eating", not "John is eating" or "John eating apple". |
| syntax_vos | APO | Identify the first and last words of each sentence, considering a sentence as a group of words that starts with a capital letter and ends with a period, question mark, or exclamation point. Ignore any punctuation, numbers, and conjunctions/prepositions at the beginning or end of the sentence. Write these two words in reverse order. If the sentence begins and ends with the same word, write it once. Treat compound words or phrases as single words. For example, 'uniroyal' and 'has' should be treated as 'uniroyal has'. |
| syntax_vos | PE2 | Identify the main subject and verb in each input sentence and form a pair. The subject is usually a noun or pronoun that the verb refers to. The verb should be the main verb of the sentence, not an auxiliary verb. For example, if the input is "The cat chased the mouse.", the output should be "cat chased". If the input is "She has eaten the cake.", the output should be "She eaten", not "She has". |
| syntax_vso | APO | Identify the main subject and the primary verb in the given sentence, regardless of their position or the complexity of the sentence. Construct a new sentence using only these two words, maintaining the order 'subject verb'. Ignore additional information, context, or implied subjects/verbs. If the subject and verb are separated by parenthetical elements, conjunctions, or other grammatical structures, still identify them as the main subject and verb. Your task is to simplify the sentence to its most basic 'subject verb' form. |
| syntax_vso | PE2 | Identify the main subject and the corresponding verb in the given sentence and construct a new short sentence using only these two words. The order should be 'subject verb'. For example, in the sentence "The dog barked at the mailman", the main subject is 'dog' and the corresponding verb is 'barked'. So, the new sentence would be "Dog barked". |

Table 17: Prompts find by prompt optimization methods. Experiments were run with 5 random data splits; In this table we report the prompts found in one run (seed=0).

| Task | Method | Prompt |
|---|---|---|
| Counterfactual Evaluation (Manual Initialization) | | |
| arithmetic_base11 | Manual Init. | You are a mathematician. Assuming that all numbers are in base-11 where the digits are 0123456789A, compute the sum of the following two numbers. |
| | APO | You are a mathematician. Assuming that all numbers are in base-11 where the digits are 0123456789A, compute the sum of the following two numbers. |
| | PE2 | You are a mathematician. Assuming that all numbers are in base-11 where the digits are 0123456789A, compute the sum of the following two numbers. |
| arithmetic_base16 | Manual Init. | You are a mathematician. Assuming that all numbers are in base-16 where the digits are 0123456789ABCDEF, compute the sum of the following two numbers. |
| | APO | You are a mathematician working with base-16 (hexadecimal) numbers. The digits are 0123456789ABCDEF, where 'A' to 'F' represent 10 to 15 respectively. Add the two given hexadecimal numbers. If the sum of two digits exceeds 15, carry the excess to the next higher digit. For instance, 'F' + '2' equals '11' in base-16, which is '1' with a carryover of '1'. The input will be two hexadecimal numbers separated by a '+'. The output should be the sum in base-16." |
| | PE2 | As a base-16 mathematician, your task is to add the provided hexadecimal numbers together. In hexadecimal system, digits go from 0 to F, with A to F representing 10 to 15 respectively. For example, to add 'B7' and '5B', convert them to decimal first: 'B7' becomes 183 and '5B' becomes 91. Their sum, 274, is '112' in hexadecimal. |
| arithmetic_base8 | Manual Init. | You are a mathematician. Assuming that all numbers are in base-8 where the digits are 01234567, compute the sum of the following two numbers. |
| | APO | You are a mathematician specializing in the octal (base-8) number system. Your task is to add two octal numbers and provide the result in octal form. In base-8, when the sum of two digits is 8 or more, you carry the value to the next higher place. For example, 7+1 in base-8 is 10. Here are some examples: |
| | PE2 | As a mathematician, your task is to add the following two numbers which are represented in base-8 (octal) format. The base-8 system uses digits from 0 to 7. Please ensure you compute the sum correctly by using base-8 arithmetic, not base-10. For example, in base-8, 7+1 equals 10, not 8. Compute the base-8 sum of these numbers, ensuring that your answer matches the provided label. For instance, if the input is "25+55", the correct output would be "102". Now, compute the base-8 sum of these numbers: |
| arithmetic_base9 | Manual Init. | You are a mathematician. Assuming that all numbers are in base-9 where the digits are 012345678, compute the sum of the following two numbers. |
| | APO | You are a mathematician working with base-9 numbers, where digits range from 0 to 8. Your task is to add two base-9 numbers. If the sum of two digits exceeds 8, carry the excess to the next higher place value, similar to base-10 arithmetic. For instance, '8+1' in base-9 equals '10'. It's crucial to interpret and present all numbers, including the final sum, in base-9. For example, if you're adding '16' and '24' in base-9, the correct sum is '41', not '40'. Now, compute the sum of the following two base-9 numbers. |
| | PE2 | You are a mathematician. Assume that all numbers you work with are in base-9, where the digits are 012345678. Your task is to add the following two numbers together, but remember to carry over any value that equals or exceeds 9 to the next digit, as is the rule when adding in base-9. For example, if you have to add 8 and 2 in base-9, the result would be 11 because 10 is not a valid number in base-9. Now, compute the sum of the following two numbers. |

Table 18: Prompts find by prompt optimization methods. Experiments were run with 5 random data splits; In this table we report the prompts found in one run (seed=0).

| Task | Method | Prompt |
|---|---|---|
| Counterfactual Evaluation (Manual Initialization) - Continued | | |
| chess_cf | Manual Init. | You are a chess player. You are playing a chess variant where the starting positions for knights and bishops are swapped. For each color, the knights are at placed that where bishops used to be and the bishops are now placed at where knights used to be. Given an opening, determine whether the opening is legal. The opening doesn't need to be a good opening. Answer "legal" if all moves are legal. Answer "illegal" if the opening violates any rules of chess. |
| | APO | You are evaluating a chess variant where knights and bishops have swapped starting positions. Knights are placed where bishops usually start, and bishops are placed where knights usually start. However, their movement rules remain the same: knights move in an L-shape and bishops move diagonally. Your task is to determine the legality of a given opening. An opening is 'legal' if all moves comply with the standard rules of chess, considering the swapped starting positions. If all moves are legal, answer 'legal'. If any move violates the chess rules, answer 'illegal'. The opening doesn't need to be a good strategy, it just needs to be legal. |
| | PE2 | You are a chess enthusiast, playing a variant of the game where knights and bishops have swapped their starting positions and movements. The knights, now placed where the bishops were, move as bishops. The bishops, positioned where knights were, move as knights. Your task is to assess the legality of a given opening, irrespective of its strategic soundness. Consider only the unique rules of this chess variant: If all moves are in accordance with these rules, your response should be "legal". However, if any move contravenes these rules, respond with "illegal". For instance, if a sequence begins with 'Bf6', it would be illegal since a bishop (moving like a knight in this variant) cannot reach 'f6' on its first move. |
| chess_original | Manual Init. | You are a chess player. Given an opening, determine whether the opening is legal. The opening doesn't need to be a good opening. Answer "legal" if all moves are legal. Answer "illegal" if the opening violates any rules of chess. |
| | APO | You are a chess expert. Given a sequence of moves, determine if they are all legal according to the rules of chess. Consider the type of piece, its legal moves, the turn order, and whether the king is put in check by its own player. If all moves are legal, answer "legal". If any move violates the rules of chess, answer "illegal". Remember, the opening doesn't need to be a good one, it just needs to follow the rules of chess. |
| | PE2 | As a chess expert, your task is to examine the given opening sequence in a chess game and determine if it adheres to the official rules of chess. Consider the sequence "legal" if every move is possible, regardless of its strategic value. However, if any move breaks a chess rule, such as moving a piece in a way it is not allowed (e.g., a knight moving like a bishop), classify the sequence as "illegal". Your response should be one of two words: "legal" or "illegal". |
| syntax_osv | Manual Init. | You are an expert in linguistics. Imagine a language that is the same as English with the only exception being that it uses the object-subject-verb order instead of the subject-verb-object order. Your task is to identify the main verb and the main subject in a sentence in this imaginary language. Show the main verb (a single word) and its subject (also a single word). |
| | APO | You are a linguistics expert. Your task is to identify the main verb and subject in a sentence of a language identical to English, but with an object-subject-verb order. The main verb is the primary action word, excluding auxiliary verbs. The main subject is the primary entity performing the action. In complex sentences, focus on the main clause. If the main subject or verb is a phrase, identify the key word that encapsulates the action or entity. If the main subject or verb is a proper noun, treat it as a single word. Your output should be a phrase consisting of the main subject and verb. For example, if the sentence is 'a milk for hispanic tastes goya concocts', your output should be 'goya concocts'. |
| | PE2 | As a linguistics expert, your task is to analyze sentences from a language that, while similar to English, employs an object-subject-verb order instead of the English subject-verb-object order. You need to identify the primary subject, who is the main entity carrying out the action, and the last verb, which is the final action described in the sentence. Output the main subject and the last verb in a single word each, and arrange them in the English order. For instance, for "apple the eats boy", your output should be "boy eats". Similarly, for sentences like "$ 4 million it will pay hunter in exchange for agreements not to compete cilcorp said", the response should be "cilcorp said", recognizing 'cilcorp' as the main subject and 'said' as the last verb. |
| syntax_ovs | Manual Init. | You are an expert in linguistics. Imagine a language that is the same as English with the only exception being that it uses the object-verb-subject order instead of the subject-verb-object order. Your task is to identify the main verb and the main subject in a sentence in this imaginary language. Show the main verb (a single word) and its subject (also a single word). |
| | APO | You are a linguistics expert analyzing a language similar to English, but with an object-verb-subject (OVS) order. Your task is to identify the main verb and the main subject in a sentence. The main verb is the primary action word, and the main subject is the primary doer of the action. They may not always be adjacent. If the main verb or subject is a compound or phrase, choose the most significant word. For sentences with auxiliary verbs, the main verb is the one conveying the primary action. After identifying, reverse the order to subject-verb for your output. For example, if the OVS order is 'apple ate John', your output should be 'John ate'. Remember, your output should always be in subject-verb order. |
| | PE2 | You are an expert in linguistics. Imagine a language that is the same as English with the only exception being that it uses the object-verb-subject order instead of the subject-verb-object order. Your task is to identify the last subject and the verb directly associated with this subject in a sentence in this imaginary language. Show the subject first (a single word) and then the verb (also a single word). For example, in the sentence "interest pay they only for 115 months , with principal payments beginning thereafter", though the last verb is "beginning", the verb directly associated with the subject "they" is "pay". Therefore, the answer is "they pay". |

Table 19: Prompts find by prompt optimization methods. Experiments were run with 5 random data splits; In this table we report the prompts found in one run (seed=0).

| Task | Method | Prompt |
|---|---|---|
| **Counterfactual Evaluation (Manual Initialization) - Continued 2** | | |
| syntax_sov | Manual Init. | You are an expert in linguistics. Imagine a language that is the same as English with the only exception being that it uses the subject-object-verb order instead of the subject-verb-object order. Your task is to identify the main verb and the main subject in a sentence in this imaginary language. Show the main verb (a single word) and its subject (also a single word). |
| | APO | You are a linguistics expert. Your task is to identify the main subject and the main verb in a sentence of an imaginary language identical to English, but with a subject-object-verb order. Your output should be in the original English order (subject-verb). Choose the most crucial word if the subject or verb is a phrase. Ignore auxiliary verbs, additional clauses, prepositional phrases, and implied words. Your output should be two single words: the main subject and the main verb. For instance, in the sentence 'John the ball threw', your output should be 'John threw'. In complex sentences, focus on the primary clause. For example, in 'that speculators a higher offer is in the wings are betting indicates', your output should be 'that indicates'. |
| | PE2 | As a linguistics expert, consider an alternate version of English that uses the subject-object-verb order instead of the traditional subject-verb-object order. Given a sentence in this alternate order, your task is to identify the main subject and the main verb and present them in the order of subject-verb. Please provide the main subject (one word) and its verb (one word) in each sentence, without considering the object. For instance, in the sentence "Jane the apple ate", "Jane" is the subject and "ate" is the verb. Therefore, the answer would be "Jane ate". |
| syntax_svo | Manual Init. | You are an expert in linguistics. Your task is to identify the main verb and the main subject in a sentence. Show the main verb (a single word) and its subject (also a single word). |
| | APO | You are a language analyst. Your task is to identify the primary subject and the primary verb in a sentence, in the order they appear. The primary subject is the main entity performing the action, and the primary verb is the main action performed by the subject. They should be part of the same clause. In complex sentences, focus on the main action and the entity performing it, considering the overall context. If there are multiple verbs or subjects, choose the pair that is most central to the sentence's meaning. Ignore conjunctions, prepositions, or other linking words that might separate the primary subject from the primary verb. If the primary subject or verb is implied, infer it from the context. Provide the primary subject and verb as a single output, with the subject first and the verb second. Both should be single words. Do not include punctuation in your output. |
| | PE2 | As a linguistics expert, your task is to determine the main verb and the main subject in a given sentence. Identify them as a single word each. The subject usually is the one performing the action, while the verb represents the action or the state of the subject. For instance, in the sentence "John plays football", 'John' is the subject, and 'plays' is the verb. Please provide the subject first, followed by the verb. |
| syntax_vos | Manual Init. | You are an expert in linguistics. Imagine a language that is the same as English with the only exception being that it uses the verb-object-subject order instead of the subject-verb-object order. Your task is to identify the main verb and the main subject in a sentence in this imaginary language. Show the main verb (a single word) and its subject (also a single word). |
| | APO | You are a linguistics expert. Your task is to identify the main verb and subject in a sentence of a language identical to English, but with verb-object-subject order. Focus on the verb and subject that carry the main action or idea. If there are multiple verbs or subjects, choose the ones that are most central to the sentence's meaning. If the verb or subject is part of a complex structure or is implied, state it explicitly. If the verb or subject is a phrase, identify the entire phrase. Your output should be in the format: 'Subject Verb'. Remember, the subject and verb may not be adjacent or single words. Use your linguistic expertise to determine the main verb and subject. |
| | PE2 | You are a linguistics expert tasked with analyzing sentences in a language similar to English but with a key difference: the order of the verb, object, and subject is changed. Your task is to identify the main subject and the first word of the verb phrase in each sentence. However, present your answer in the subject-verb-object order commonly used in English. In other words, reveal the main subject (a single word) followed by the first word of the verb phrase (also a single word). For example, if the sentence is "continue to lead gold stocks and utilities , may signal that is the market in for rough times it", your answer should be "it signal". |
| syntax_vso | Manual Init. | You are an expert in linguistics. Imagine a language that is the same as English with the only exception being that it uses the verb-subject-object order instead of the subject-verb-object order. Your task is to identify the main verb and the main subject in a sentence in this imaginary language. Show the main verb (a single word) and its subject (also a single word). |
| | APO | You are a language expert analyzing a unique language similar to English, but with verb-subject-object order. Your task is to identify the main verb and subject in a sentence. The main verb is the key action, and the main subject is who or what is doing this action. In complex sentences, focus on the most important action. If multiple verbs or subjects exist, choose the most central to the sentence's meaning. Treat auxiliary or compound verbs as one unit with their main verb. Your output should be the main subject followed by the main verb (both as single words)." |
| | PE2 | As a linguistics expert, consider an alternative English language that uses verb-subject-object order instead of the standard subject-verb-object order. Your task is to identify the main subject and the main verb in a sentence in this imaginary language. Display the main subject (a single word) followed by its verb (also a single word). For instance, if the input is "compares that with 3.5 % butterfat for whole milk", the output should be "that compares". Similarly, for "believe they is technology one of their best bets", the output should be "they believe". |

