# OpenReview forum: "Prompt Engineering a Prompt Engineer"
_ICLR.cc/2024/Conference — Submitted to ICLR 2024_

### Official Review · Reviewer_MVYf · 2023-10-31

**Soundness:** 2 fair
**Presentation:** 2 fair
**Contribution:** 2 fair
**Rating:** 3
**Confidence:** 4

**Summary:**

Optimizing prompts for LLMs is challenging, but crucial. In this work, the authors studied and proposed an automatic method to construct meta-prompts for new prompt proposal so that generated and edited prompts could be used to guide LLMs to perform better. They analyzed and investigated key components to build meta-prompt, such as providing step-by-step detailed instructions and context (see Sec 5.1 for empirical investigation). They also combined concepts in optimizers and used a gradient-based approach to refine prompts. In their experiments, they included four tasks and three existing baseline works to evaluate their proposed methods. The main results showed that PE2 approach can improve baseline performances. In addition, PE2 generates more high-quality prompts and specific prompt edits.

**Strengths:**

* Originality: This paper proposed that to achieve a helpful meta-prompt we should enrich the meta-prompt with additional instructions and context. Standing on this, they developed several components to provide detailed instruction and context to prompt proposal LLM.
* Quality: The experimental results look promising and perform better than the existing two automatic prompt optimization methods.
* Significance: Prompt engineering is important to maximize the utility of LLMs. Instead of crafting prompts by human, this work proposed an automatic approach to leverage other LLMs to generate new prompts for downstream tasks. The numbers in their results validated the effectiveness of their proposed method.

**Weaknesses:**

* The clarity of the way to update hard prompt with gradient-based optimizer can be enhanced and improved by providing details, especially how do we access the gradients to help LLM refine hard prompts?
* The iteration for the optimizer is set to 3 in the experimental setup. How does this come from and be enough to optimize prompts? This part is not well-supported.
* In Figure 3 and Figure 4, I am not sure why are we doing comparison across different training timestamps. Instead, should we focus on the final timestamp or the converged step to confirm it's finalized and optimized for any further investigations? This comparison to display the dynamics is confusing me.

**Questions:**

* The footnote #3 in page 4 is not well-supported and is confusing. How does the analogy come from and be translated?
* In paragraph "Incorporating Concepts in Optimizers", what's the concept here? Is there any formal definition about concept?
* (minor comment) In Figure 1, the results can be improved by adding variance/range in the accuracy performance. It helps to enhance the soundness of this work.

---

> ### Author Response · Authors · 2023-11-22
>
> Thank you so much for your review! After reading your review we find that there might be some confusion right now. We hope our response can resolve some of them and help you reassess our work.
>
> ### Clarification
>
> > In this work, the authors studied and proposed an automatic method to construct meta-prompts …
>
> * We work on an automatic method to construct prompts (NOT meta-prompts).
> * In this work, prompts refer to instructions used directly in the task, e.g., “Let’s think step by step.” Constructing the prompts is automatic.
> * Meta-prompt refers to the instructions for refining prompts, e.g., “inspect the current prompt and a batch of examples, then propose a new prompt.” Constructing the meta-prompt is done manually by us in this paper.
>
> > They also combined concepts in optimizers and used a gradient-based approach to refine prompts.
>
> * We want to clarify that __our method is NOT a gradient-based optimization method__. Our method and all baseline methods in this work are operating on textual prompts, using the reasoning and text processing capabilities of LLMs and using __only forward calls of LLMs__.
>
> * Here is a minimal example:
>   * Old prompt: “Let’s think step by step.”
>   * LLM-generated feedback (analogous to “gradients”): “The prompt should be edited to guide the model to perform subtraction.”
>   * LLM-generated new prompt: “Let’s solve this problem step by step. Remember to add or subtract as needed.”
> The prompt refinement process is done by calling LLMs directly, instead of performing actual gradient descent on LLM parameters.
>
> * Regarding the optimization-inspired components, they are not actually operating on the LLM parameters or gradients. Instead, they are all verbalized in language instructions. For example, step size means we use “change up to 10 words” when instructing the model to generate a new prompt.
>
> ### Response to weaknesses
>
> > The clarity of the way to update hard prompt with gradient-based optimizer can be enhanced and improved by providing details, especially how do we access the gradients to help LLM refine hard prompts?
>
> * As mentioned in the example above, our method is not a gradient-based optimizer. The textual feedback generated by the model is used to steer the prompt editing process, and therefore is analogous to “gradients.” We hope the example above helps clarify this.
>
> > The iteration for the optimizer is set to 3 in the experimental setup. How does this come from and be enough to optimize prompts? This part is not well-supported.
>
> * We set the number of prompt edits to be 3 due to cost concerns. Empirically, performance tends to plateau after 3 steps.
> For each prompt candidate, we need to run it on the training set to find the model errors and run it on the dev set for validation and prompt selection. This leads to a significant number of LLM calls.
>
> > In Figure 3 and Figure 4, I am not sure why are we doing comparison across different training timestamps. Instead, should we focus on the final timestamp or the converged step to confirm it's finalized and optimized for any further investigations? This comparison to display the dynamics is confusing me.
>
> * We agree with you that the prompt on final/best prompts are most important. Our table 1 and table 2 are dedicated to results on the final prompt. (Now they are table 3 and table 1 in the updated paper)
> * Figure 3 and 4 illustrates the _accuracy distribution_ of the proposed prompts over the course of prompt engineering. We create them for some additional analysis and they indeed provide valuable insights. For example, in Figure 3 (now Figure 4 after paper update) we conducted an ablation study on the meta-prompt components and we found that by removing these components, prompt quality at each timestamp is lowered.
>
> ### Response to questions
>
> > The footnote #3 in page 4 is not well-supported and is confusing. How does the analogy come from and be translated?
>
> * Please refer to our clarification section above. We believe it is helpful in answering this question.
> * The textual feedback generated by inspecting a batch of model errors, and therefore this is analogous to “loss.backward()”.
> * Generating a new prompt based on the old prompt and the textual feedback, is analogous to “optimizer.step()”.
>
> > In paragraph "Incorporating Concepts in Optimizers", what's the concept here? Is there any formal definition about concept?
>
> * The concepts are batch size, step size and momentum that are commonly used in gradient-based optimization. The bulletin points in that paragraph provide more detail on each of them.
> * Given feedback from reviewer x9xJ and reviewer uHAL, we agree this is a little confusing and we decide to more this part to the appendix in our next paper update.
>
> > Adding variance/range in Figure 1
>
> * Thank you for raising this! We will do this whenever applicable. For bars presenting the average across multiple tasks, there is not a straightforward way to compute variance and thus we cannot include them here.

---

### Official Review · Reviewer_iQjD · 2023-11-01

**Soundness:** 4 excellent
**Presentation:** 4 excellent
**Contribution:** 4 excellent
**Rating:** 8
**Confidence:** 4

**Summary:**

The paper presents PE2, a prompt optimization method that leverages a well-designed meta-prompt to iteratively improve on proposal prompts when presented with examples of the task at hand. The meta-prompt in particular leverages two-step task description, step-by-step reasoning, and context specification. A prompt engineering tutorial is further explored but ultimately discarded due to its length and inefficacy. Furthermore, optimisation-based concepts are introduced such as batch size, learning rate, and momentum although the latter two are not used in the final PE2 due to lack of consistent empirical improvements. PE2 is evaluated on various mathematical reasoning, instruction induction, and counterfactual evaluation tasks.

**Strengths:**

- PE2 is well-designed and the prompt choices made are justified both empirically and using real-life examples.
- Experimental results demonstrate that components in the final PE2 contribute to improvements in accuracy. Furthermore, various other aspects such as prompt tutorial and use of momentum are also evaluated empirically before being excluded in the final method.
- Experiments are extensive and explore various aspects of the meta-prompt including the ability to handle poor initializations, reasoning capabilities, and poor performance arising from hallucinations and ignoring instructions.
- Overall, the paper is very well-written and easy to follow.

**Weaknesses:**

- The evaluation uses Text-Davinci-003 as the main model but GPT4 when handling the meta-prompt, prompt evaluation and update proposals. It's noted that for two baselines that used the Text-Davinci-002, the results are recreated using Text-Davinci-003. It begs the question of whether APO or APE with GPT4 paraphrases could potentially perform better than reported. Was GPT4 also used in any of the baselines? Have the authors evaluated using Text-Davinci-003 to also handle the meta-prompt? Given that hallucinations hurt performance considerably it would be important to answer these questions for a fair comparison to baselines.
- Text-Davinci-003 further lacks some capabilities of GPT3.5 and GPT4. Although I am sympathetic to the cost constraints, I would find it valuable if the authors had any existing experimental results where either of these models is used as the main model to optimize for (even if the results are not fully run across all benchmarks).
- I suspect that the authors will agree that as a result, one could argue that the meta-prompt itself could be promptly optimized with PE2. Although they allude to this at the very end, I believe that a more detailed discussion on this front could be useful.

**Questions:**

Please answer the questions noted above. Overall, strong submission with several simple but empirically powerful contributions that enable improved prompt optimization. Happy to recommend acceptance.

---

> ### Author Response · Authors · 2023-11-22
>
> Thank you so much for your review! It’s very encouraging to read your review. Below are our responses.
>
> > Was GPT4 also used in any of the baselines? Have the authors evaluated using Text-Davinci-003 to also handle the meta-prompt?
>
> * Yes. Our experiments on APO and Iterative APE are using GPT-4 as the prompt proposal model. We make sure the experiments are controlled and __the meta-prompt is the only changing variable__.
> * In the initial stage of this project we tried using GPT-3.5-turbo as the prompt proposal model. The overall impression we have is that GPT-3.5-turbo is sometimes confused at the concept of prompt (task description) and the task input and cannot distinguish them. So we decide to proceed with GPT-4
>
> > Text-Davinci-003 further lacks some capabilities of GPT3.5 and GPT4. Although I am sympathetic to the cost constraints, I would find it valuable if the authors had any existing experimental results where either of these models is used as the main model to optimize for (even if the results are not fully run across all benchmarks).
>
> * We decide to use instruct models to be consistent with prior work. Unfortunately we didn’t use GPT 3.5/4 models as the task mode because they are chat completion models.
>
> > I suspect that the authors will agree that as a result, one could argue that the meta-prompt itself could be promptly optimized with PE2. Although they allude to this at the very end, I believe that a more detailed discussion on this front could be useful.
>
> * Yes, we are very excited about this potential future direction!!
> * Conceptually, we can replace $p^{(t)}$ and $p^{(t+1)}$ in Eq (2) with the meta-prompt $p^{meta}$ directly. However, we think three challenges (and broader questions) arise if we pursue this method:
>   * How to collect data for such a study? To ensure this meta-prompt is universal we may need a large collection of tasks along with prompt optimization history associated with them. Creating a resource like this will be a large effort.
>   * How to automatically optimize the meta-prompt when there are no ground truth labels for prompt engineering? Math problems have ground-truth answers so that PE2 can reason on them. The task of prompt engineering does not have ground truth labels, and this makes the meta-prompt optimization process more fuzzy.
>   * It would be very costly to run and even evaluate a system like this. To evaluate one $p^{meta}$ candidate, we will need to use it for prompt optimization on various tasks as evaluation. We would expect the “learning” process of the meta-prompt to be a magnitude more costly.
> * We noticed two really exciting recent works related to this direction archived in the past two months: https://arxiv.org/abs/2309.16797 and https://arxiv.org/abs/2310.02304

---

### Official Review · Reviewer_uHAL · 2023-11-01

**Soundness:** 3 good
**Presentation:** 2 fair
**Contribution:** 2 fair
**Rating:** 5
**Confidence:** 3

**Summary:**

The paper proposes an approach (PE2) to automatically improve the prompt used in LLMs. The key idea is to hand-design a better meta-prompt (prompt to generate better prompts). The improved meta prompt is then used in the inner loop of an iterative search process constructed over the space of prompts using a dataset of task examples to guide the search for better prompts. Experiments are performed on a variety of tasks. The results indicate that the proposed method outperforms existing automated prompt engineering techniques and one human-engineered prompts.

**Strengths:**

+ The paper studies an important problem of automatically improving the input prompt to LLMs. Progress here is likely to be of wide interest to the community.

+ The primary contributions of the paper are algorithmic and empirical. The main contribution is the final version of the hand-engineered meta prompt used in PE2. Experiments to study its empirical performance indicate it performs well compared to baseline prompt generators, both automated and human.

+ The illustrative examples are useful to quickly grasp the main ideas being proposed. While some important implementation details are a bit difficult to follow, the appendixes contain sufficient information to mostly fill in the blanks.

**Weaknesses:**

- The algorithmic contribution seems a bit thin. While this might be expected given the black box trial-and-error nature of prompt engineering, there is very little by way of novelty beyond the meta prompt design itself. The issue of limited novelty might be alleviated with additional insight about algorithmic components. For example, why does performance plateau quickly as $t$ increases? What kinds of prompts are generated at the end of long searches?

- The current presentation makes it hard to "separate the wheat from the chaff". I found it challenging to quickly identify the "final" best variant of the meta-prompt. The paper describes a number of components, but discards some in the final version constituting PE2 (used to generate Figure 1). If I've understood correctly, discarded items include the prompt tutorial, step size and momentum. If true, perhaps the main paper might be simplified to only describe what is actually used in PE2 / Figure 1 with  supporting evidence with the rest moved to the appendix as negative experiments. Just a suggestion.

**Questions:**

- How exactly are the examples in Line 42 in B.4 selected? It seems like PE2 uses 2 negative examples from D_train ("hard negative sampling"). Is this correct? Does batch_size refer to the total number of examples or just the negative examples?

- Appendix C.1.1 suggests including more in-context examples (e.g., 3, 10) improves performance. If so, why is batch size set to 2 in PE2?

- Where is $D_\text{dev}$ (validation dataset) used in Algorithm 1? (Perhaps in Select-Best?)

- Is the use of the optimization terminology beneficial? The term "batch" in the context of LLMs is reasonably well understood to refer to the set of examples used during fine-tuning and much less so to refer to the set of input-output examples in the prompt. Since step size and momentum don't seem to be anyway used in the final PE2 version, might it be clearer to simply describe the meta-prompt in its final form without reusing popular, well-understood terminology from optimization?

- Do you have any insight into why prompt performance reaches a plateau so quickly (by $t$ = 3)? How much performance gain would be lost wrt Figure 1 if only a single round of improvement (t = 1) was conducted?

- What, if any, effect does the choice of examples ("batch") have on performance? Is there a notion of "active learning" that might be worth incorporating into the meta-prompt? This is a complete reformulation of the optimization problem so probably outside the scope of the paper. I was wondering if you had any empirical insights here as it seems related to the primary objective of finding the prompt producing best performance wrt the tasks using an LLM.

---

> ### Author Response · Authors · 2023-11-22
>
> Thank you for your thorough and detailed review! Please find our responses below.
>
> > The algorithmic contribution seems a bit thin. … The issue of limited novelty might be alleviated with additional insight about algorithmic components. …
>
> * In response to “Why does performance plateau quickly as t increases?”: Our hypothesis is that “let’s think step by step” is already a near-optimal prompt and thus the performance plateaus quickly when using it as the initialization. According to Figure 4(b) in the paper (now Figure 3(b) in the updated paper), when alternative initializations are used, PE2 is able to improve the performance gradually and the performance hasn’t plateaued at t=3.
> * In response to “What kinds of prompts are generated at the end of long searches?”: We summarized some notable prompt edits in Table 4, 5, 8 (Now Table 4, 5, 9 in the updated version). PE2 is able to correct wrong or incomplete task instructions, provide more specific context and details. In our upcoming paper update, we will show that PE2 can lay out multi-step plans in the prompt.
>
> > The current presentation makes it hard to "separate the wheat from the chaff".
>
> * Thank you for raising this point! This is also pointed out by reviewer x9xJ and thus we will update our paper by moving the meta-prompt components with mixed observations to the appendix. We will push an update to the paper very soon.
>
> > How exactly are the examples in Line 42 in B.4 selected? It seems like PE2 uses 2 negative examples from D_train ("hard negative sampling"). Is this correct? Does batch_size refer to the total number of examples or just the negative examples?
>
> * Yes, by default we _only_ select examples in D_train that the model produces _wrong_ answers. In this case batch_size = # negative examples = # selected examples.
> * We tried selecting random examples in D_train instead of negative examples, and this is denoted as “- hard negative sampling” in Table 1 (now Table 3 in the updated paper). Performance is slightly worse when using random examples.
>
> > Appendix C.1.1 suggests including more in-context examples (e.g., 3, 10) improves performance. If so, why is batch size set to 2 in PE2?
>
> * Apologies for the confusion! After the main experiments on academic benchmarks are done, we decide to try out our method on an industrial prompt. The industrial prompt is substantially longer (5k+ tokens), which creates a very different optimization space. The default hyperparameters on academic benchmarks do not work directly. This necessitates a separate hyperparameter study, which we described in Appendix C.  Furthermore, to the best of our knowledge this is the first work to look at directly optimizing very large prompts deployed application prompts.
>
> > Where is  (validation dataset) used in Algorithm 1? (Perhaps in Select-Best?)
>
> * Yes you’re correct! Select-Best is based on D_dev. We will make this clearer in our updated paper.
>
> > Is the use of the optimization terminology beneficial?
>
> * As mentioned earlier, after reading the reviews we realized that this causes confusion and we will push an update to the paper. Thank you for the suggestion!

---

> ### Author Response · Authors · 2023-11-22
>
> > Do you have any insight into why prompt performance reaches a plateau so quickly (by t = 3)? How much performance gain would be lost wrt Figure 1 if only a single round of improvement (t = 1) was conducted?
>
> * As discussed above, we believe the performance plateaued quickly because “Let’s think step by step” is a very strong initialization prompt. The optimization trajectory will be different when alternative initializations are used (Figure 4(b); now Figure 3(b) after paper update). In general, we think the answer to your question is dependent on the task and the initialization prompt.
> * During the response period, we tested the prompts found at t=1 and t=2 for three tasks: Multiarith, Date Understanding, and Movie Recommendation. (For these three tasks, the best overall prompts are all found at t=2). We hope the results below answer your question "How much performance gain would be lost if a single round was conducted".
> * MultiArith
>   * t=0, “Let's think step by step.”, Dev Acc 85.0, Test Acc 86.0
>   * t=1, “Let's solve the problem step by step. First, identify all the numbers and what they represent. Then, perform the necessary calculations to find the answer.”, Dev Acc 87.0, Test Acc 86.8
>   * t=2, “Let's solve this problem by considering all the details. Pay attention to each piece of information, remember to add or subtract as needed, and perform the calculations step by step.”, Dev Acc 92.0, Test Acc 92.3
> * Date Understanding
>   * t=0, “Let's think step by step.”, Dev Acc 43.0, Test Acc 39.1
>   * t=1, “Let's carefully analyze the information and perform calculations if necessary, step by step.”, Dec Acc 46.0, Test Acc, 43.2
>   * t=2, “Analyzing the given information, let's calculate the solution. Remember to consider the context provided, such as references to 'today' or specific dates.”, Dev Acc 54.0, Test Acc 54.4
> * Movie Recommendation
>   * t=0, “Let's think step by step.”, Dev Acc 58.0, Test Acc 57.0
>   * t=1, “Consider the genre, plot, and style of the input movies. Using this information, think step by step to identify which of the following options is most similar to the given movies.”, Dev Acc 74.0, Test Acc 78.0
>   * t=2, “Considering factors such as genre, director, actors, release period, audience target, animation style, and humor, analyze the similarities among the given movies and identify the movie from the options that shares the most similarities.”, Dev Acc 82.0, Test Acc 79.0
>
> > What, if any, effect does the choice of examples ("batch") have on performance? Is there a notion of "active learning" that might be worth incorporating into the meta-prompt?
>
> * Thank you for bringing this up! It will be absolutely interesting to study this.
> * For now we tried comparing a batch of _model failure examples_ and a batch of _random examples_, and found that model failures are slightly more useful (see Table 1 “- hard negative sampling”)(Now it's Table 3 in the updated paper).
> * Our intuition is that, similar to how the choice of examples are important to the performance of in-context learning, the choice of examples for generating feedback will be important to prompt optimization.
> * Some of our initial thoughts towards this direction: It is straightforward to identify highly educational failures for math problems. If the number is off by a lot, it suggests the failure is more fatal and more educational. However it is less clear how to actively select the most valuable examples for non-math tasks. Perhaps model confidence information will be helpful. It will be interesting to get inspiration from active learning literature too.

---

### Official Review · Reviewer_x9xJ · 2023-11-04

**Soundness:** 2 fair
**Presentation:** 2 fair
**Contribution:** 2 fair
**Rating:** 3
**Confidence:** 3

**Summary:**

The paper proposed an approach to propose automate design for the instructions in the prompt. The approach consists of prompt initialization, new prompt proposal, search procedure. The proposed approach includes a few tricks, including Providing Detailed Instruction and Context (prompt engineering tutorial, Two-step Task Description. Step-by-step Reasoning Template, Context Specification), Incorporating concepts in optimizers (batch size, step size, history and monentum,  back-tracking
hard negative sampling). The paper improves improvement over  other methods like APO, APE.

**Strengths:**

The paper works on an interesting and important problem.  The work includes a few interesting tricks, such as batch size, step size, etc， which is analogous to optimization. The paper provides good ablation study.

**Weaknesses:**

1 The effectiveness of the proposed approach is not conclusive.
- First, although the approach includes a few interesting tricks, the effectiveness of them are unclear, as indicated by ablation study

"The optimizer-inspired concepts can improve the performance occasionally, but the current experiments do not give a definitive conclusion regarding their utilities"; "We do not observe significant improvement by incorporating prompt engineering tutorial."

While the readers appreciate the author's honesty and agree negative results are still informative, it will be good the author explores more on which scenarios the proposed tricks are more likely to be helpful.

- Second, most of the analysis and ablation studies (Table 1,2,3) are on simple math datasets MultiArith, GSM8K. Are the proposed approach work on harder math datasets (i.e. https://paperswithcode.com/dataset/math), which are more close to real-world usage? While the paper also evaluate on "instruction induction" and "counterfactual eval" (Figure 1), the approach still haven't tested on the more representative tasks categories (i.e. QA, test summarization, etc) to be persuasive.  Automate prompt design approach should aim work on general situations. Does the approach work on a more general use case?

Also,  for GSM8K, no the SOTA is above 90. While we understand the authors are using a less strong foundation models, the big gap between the sota still draw concerns on the effective of the methods. Does it work on better foundation models?

- Third, some recent papers sharing about prompt design is also interesting. What are the proposed methods compared with these methods? meta prompt optimization section 4.2 of https://arxiv.org/pdf/2309.03409.pdf. prompt design to optimize demonstrations: https://arxiv.org/abs/2305.14106

**Questions:**

See above

---

> ### Author Response · Authors · 2023-11-22
>
> Thank you for your feedback on our paper! We hope the responses below can help address your concerns.
>
> > First, although the approach includes a few interesting tricks, the effectiveness of them are unclear, as indicated by ablation study.
>
> * Sorry about the confusion! To clarify, our ablation study shows that __three components we propose (two-step task instruction, context specification and step-by-step reasoning template) are indeed effective__ (Table 1 and Figure 3)(They are now Table 3 and Figure 4 in the updated paper). We believe these are important contributions in our paper. PE2 powered by these components also outperform prior methods such as Iterative APE and APO on a wide range of tasks (Figure 1).
> * We agree that results on optimizer-inspired components such as optimization history and step size are currently mixed. As pointed out by reviewer uHAL, we are re-organizing the paper to highlight the effective components and defer the remaining mixed results in the appendix. Please expect an updated version of the paper before the discussion period ends.
>
> > Second, most of the analysis and ablation studies (Table 1,2,3) are on simple math datasets MultiArith, GSM8K. Are the proposed approach work on harder math datasets?
>
> * We choose MultiArith and GSM8K as the main testbed by following past work on (manual/automatic) prompt engineering (https://arxiv.org/abs/2205.11916; https://arxiv.org/abs/2211.01910). Concurrent works (https://arxiv.org/abs/2309.03409; https://arxiv.org/abs/2309.16797) also use these two datasets. We believe it is reasonable to focus our analysis on these two datasets.
> * We agree with you that it will be very interesting to apply these methods to harder datasets! However due to resource constraints we are unable to further explore this.
>
> > Also, for GSM8K, now the SOTA is above 90. While we understand the authors are using a less strong foundation models, the big gap between the sota still draw concerns on the effective of the methods.
>
> * Our experiments on GSM8K is about finding the optimal prompt in the __zero-shot chain-of-thought__ setting, as done in https://arxiv.org/abs/2205.11916.
> * We learned that the current state-of-the-art performance on GSM8K is based on __few-shot chain-of-thought prompting__ or __additional human-designed verification workflow__. We believe these advances are orthogonal to us and the results are not directly comparable.
> * While we agree pushing forward state-of-the-art performance is important, we would like to highlight that __our focus is enabling LLMs to improve LLM performance automatically__. We really hope one day LLMs can conduct research and push forward state-of-the-art by themselves one day (https://arxiv.org/abs/2310.03302).
>
> > The approach still haven't tested on the more representative tasks categories (i.e. QA, test summarization, etc) to be persuasive.
>
> * We agree with you that generality is important in evaluating automatic prompt engineering methods!
> * We hope our experiments on an industrial prompt (Figure 1) helps address your concern regarding the generality of our approach. The underlying task is a 3-level hierarchical classification task on domain and intent classification.
> * During the rebuttal period we applied PE2 to two tasks from BIG-bench Hard (https://arxiv.org/abs/2210.09261): date understanding and movie recommendation. We believe these two tasks are close to everyday scenarios. Results suggest that PE2 is effective on these tasks. We will include these results in the upcoming paper update.
>
> |      | Date Understanding | Movie Recommendation|
> | :--- | :----: | :----: |
> | Initialization      |  39.1   | 57.0 |
> | Iterative APE |  46.7 |  67.3 |
> | APO | 45.0 | 75.0 |
> | PE2 | 54.0 | 79.0 |
>
> * We decided _not_ to test on common language tasks such as QA and summarization, as modern instruction-tuned models are already trained to do QA and summarization. Prompt engineering is most effective for customized and ad-hoc tasks unseen during model training.

---

> > ### Author Response · Authors · 2023-11-22
> >
> > > Third, some recent papers sharing about prompt design are also interesting. What are the proposed methods compared with these methods?
> >
> > Thank you very much for bringing these works to our attention!
> >
> > Regarding OPRO (https://arxiv.org/abs/2309.03409):
> > * Firstly, please kindly note that OPRO was archived three weeks before ICLR submission deadline. We request that the novelty and contribution of our work is evaluated independently.
> > * In our submitted version, we discussed our differences and thoughts on OPRO in Appendix A.1. We think OPRO and PE2 represent two very different optimization strategies. OPRO mainly uses prompt accuracy as reward signal and asks for _implicit_ optimization; PE2 mainly uses model failure cases for generating _explicit_ feedback.
> > * As mentioned in the OPRO paper, “We consider incorporating explicit natural language feedback on generated solutions for later optimization steps as _future work_.” PE2 can be considered as initial steps towards this direction.
> >
> > Regarding Self-Adaptive Prompting (https://arxiv.org/abs/2305.14106):
> > * Thank you for bringing this work to our attention! It’s a cool idea to automatically construct in-context demonstrations based on consistency, diversity and repetition. We will include it in the related work section in our upcoming update.
> > * In our study we limit our scope to prompt engineering the task descriptions (see our footnote 1). Following prior work (https://arxiv.org/abs/2205.11916), the prompts are limited to having less than 50 tokens in our experiments. Hence the two problems settings are not consistent and we cannot make direct comparison.
> > * It will be interesting to study the synergistic effect of self-adaptive prompting and our PE2 method, which we leave as future work.

---

### Author Response · Authors · 2023-11-23

Dear reviewers,

We are encouraged that you find our research topic important (x9xJ, uHAL), of interest to the research community (uHAL); our work contains well-designed, extensive experiments (iQjD) and the results are promising (MVYf).

Thank you all for your feedback (“gradients”)! We have incorporated them into our paper (“prompt”) and just pushed an update to our paper. Here is a summary of main updates we made to the paper:

* __Re-organize the paper to highlight useful meta-prompt components.__ Per reviewer x9xJ and uHAL’s suggestion, and reviewer MVYf’s feedback, we decide to highlight the three useful meta-prompt components in the main paper, and defer the remaining inconclusive components to Appendix A.2. We also re-order the subsections in section 5. Section 5.1 has the main results, Section 5.2 has the ablation study, Section 5.3 has additional analysis and case study.
* __New experiments on two tasks representing real-world applications.__ To address reviewer x9xJ’s concern about the generality of PE2, we apply it to date understanding and movie recommendation, two challenging tasks from BIG-bench Hard that are closely related to everyday scenarios. On these two tasks PE2 consistently outperforms baseline methods. Full results are in Table 8 in the updated paper.
* __New observations on the two new tasks.__ We find that PE2 is able to devise multi-step plans for complex tasks when doing prompt engineering. For movie recommendation, PE2 automatically proposes aspects to consider (e.g., genre, plot, style). For date understanding, PE2 suggests that it is important to first figure out “what day is today”. Please check out Table 4 for more details. We are very excited about these new observations!

Special thanks to reviewer uHAL for bringing up active learning and reviewer iQjD for bringing up using the meta-prompt to optimize itself. These discussions opened up future directions and we enjoyed the brainstorming process.

Please let us know if you have follow-up questions or additional comments!

Authors of submission 6421

---

### Meta-Review · Area_Chair_24ED · 2023-12-12

**Metareview:**

This paper explores the timely problem of automated prompt engineering/optimization. The authors propose the algorithm PE2 where the idea is designing a meta-prompt which can iteratively refine the initial prompt based on task description and failure cases. Overall, the reviewers found the paper to be well-motivated and outcomes to be promising. I believe the authors put a genuine effort and provide a thorough study and ablation experiments including production prompt setting. However the conclusion is that, this work suffers from a few issues and is not ready for publication at this point. To summarize some of the points raised by reviewers:

(1)	The novelty of the proposed method does not come across. This is partly because there are earlier works with similar flavor. However, the bigger issue seems to be that it is difficult to identify what the 2-3 key novelties compared to prior art that underlie the success.

(2)	Some of the evaluations are not SOTA, especially GSM8K. I think the paper will benefit from at least some evals where the authors use GPT-4 as the base model to assess PE2’s benefits under a more optimized setting.

(3)	Some reviewers and I found the “optimization” terminology a bit misleading. For instance, one of the reviewers thought the optimization is actually gradient-based whereas in fact it is a blackbox LLM-based optimization. While the analogy to continuous optimization is nice and insightful, it might be good to ensure accurate phrasing and avoid confusion. Perhaps some of the terminology can be better phrased through zeroth-order (blackbox) or evolutionary optimization.

(4)	The authors use very few steps of optimization (like 3 iterations). This should be further motivated. Perhaps the benefit of more iteration saturates due to inability to select good examples to improve the performance (as pointed out by one of the reviewers i.e. active learning aspect).

Overall, I believe this will be a stronger and more insightful paper once authors incorporate reviewer suggestions.

**Justification For Why Not Higher Score:**

N/A

**Justification For Why Not Lower Score:**

N/A

---

### Decision · Program_Chairs · 2024-01-16

Reject